# Low-dose penicillin in early life induces long-term changes in murine gut microbiota, brain cytokines and behavior

Sophie Leclercq[1,2], Firoz M. Mian[1], Andrew M. Stanisz[1], Laure B. Bindels[3], Emmanuel Cambier[4], Hila Ben-Amram[5], Omry Koren[5], Paul Forsythe[1,6] & John Bienenstock[1,2]

There is increasing concern about potential long-term effects of antibiotics on children's health. Epidemiological studies have revealed that early-life antibiotic exposure can increase the risk of developing immune and metabolic diseases, and rodent studies have shown that administration of high doses of antibiotics has long-term effects on brain neurochemistry and behaviour. Here we investigate whether low-dose penicillin in late pregnancy and early postnatal life induces long-term effects in the offspring of mice. We find that penicillin has lasting effects in both sexes on gut microbiota, increases cytokine expression in frontal cortex, modifies blood–brain barrier integrity and alters behaviour. The antibiotic-treated mice exhibit impaired anxiety-like and social behaviours, and display aggression. Concurrent supplementation with *Lactobacillus rhamnosus* JB-1 prevents some of these alterations. These results warrant further studies on the potential role of early-life antibiotic use in the development of neuropsychiatric disorders, and the possible attenuation of these by beneficial bacteria.

[1] McMaster Brain-Body Institute at St Joseph's Healthcare Hamilton, 50 Charlton Avenue East T3304, Hamilton, Ontario, Canada L8N 4A6. [2] Department of Pathology and Molecular Medicine, McMaster University, 50 Charlton Avenue East, Hamilton, Ontario, Canada L8N 4A6. [3] Metabolism and Nutrition Research Group, Louvain Drug Research Institute, Université Catholique de Louvain, Avenue E. Mounier 73, Brussels 1200, Belgium. [4] Faculty of Medicine, Université Catholique de Louvain, Brussels 1200, Belgium. [5] Faculty of Medicine, Bar-Ilan University, Henrietta Szold 8, Safed 1311502, Israel. [6] Firestone Institute for Respiratory Health and Department of Medicine, McMaster University, 50 Charlton Avenue East, Hamilton, Ontario, Canada L8N 4A6. Correspondence and requests for materials should be addressed to S.L. (email: leclercs@mcmaster.ca) or to J.B. (email: bienens@mcmaster.ca).

Oral antibiotics (AB), particularly the penicillins, are the most frequently dispensed drugs in children worldwide[1]. There is increasing concern that AB exposure early in life may have long-term detrimental consequences for health[2]. Epidemiological studies report an association between the use of AB during the perinatal period and an increased risk of developing childhood diseases that may persist into adulthood. For instance, maternal use of AB during pregnancy or breastfeeding is a risk factor for development of wheezing and allergy in the offspring[3,4], and AB exposure during the first years of life is associated, dose-dependently, with allergic diseases[5,6], inflammatory bowel diseases[7], obesity[8,9], as well as poorer neurocognitive outcomes later in life[10]. Recent experimental studies[11,12] have found that alteration of the gut microbiota induced by early life AB exposure may drive lasting immune and metabolic consequences in mice. More particularly, Russell et al.[13] showed in mice that the pre-weaning period is critical for antibiotic-driven shift in microbiota to alter the immune response and increase susceptibility to allergy[11]. In addition, Cox et al.[12] showed that mice treated continuously with low-dose penicillin from 1 week before birth until weaning, exhibited higher body weight and fat mass in adulthood, although the microbial structure returned to normal after 4 weeks of AB cessation. These results strongly suggest that early-life dysbiosis can have long-term detrimental health effects.

There is now mounting evidence, in humans and rodents, for the role of specific microbial compositions in modulating brain function including behaviour[14–17]. Complete absence of intestinal bacteria (in germ-free mice) results in modification of blood–brain barrier (BBB) permeability[18], impaired immune response of the microglia[19], increased myelination[20], hyperactivity of the hypothalamus–pituitary–adrenal axis[21], changes in brain neurochemistry[22] and decreased anxiety and social behaviours[22,23]. Germ-free mice provide useful models to establish possible causality in rodent gut microbiota–brain interactions, but provide only suggestive clinical relevance. Several studies have shown that high doses of a cocktail of AB, including anti-fungal agents, in adult or adolescent mice induced changes in gut microbiota associated with behavioural

alterations[24–27], but these combinations of AB are never routinely used in clinical practice. By contrast, probiotics administration in mice restores intestinal barrier function[28], normal stress response[21] and brain chemistry[29] and, in humans, changes brain activity[30]. In addition, Lactobacillus rhamnosus JB-1 have demonstrated psychoactive and neuroactive properties[31], by reducing anxiety and depression-like behaviours in healthy mice via the vagus nerve[32].

Here we investigate the long-term consequences of a low dose of penicillin given to mice during the perinatal period (from 1 week before birth to weaning) on gut microbiota, intestinal barrier function, BBB integrity, cytokines expression and behaviour. Because mood disorders occur more frequently in women than men[33], we looked for differential effects in both male and female mice. We also tested whether concurrent supplementation with Lactobacillus rhamnosus JB-1 (JB-1) may counteract the biological and behavioural changes induced by early life AB. We find that AB-treated mice have lasting changes in gut microbiota, modified BBB integrity in the hippocampus, increased levels of cytokines in the frontal cortex and behavioural alterations including decreased anxiety-like behaviour and increased aggression in males as well as reduced social behaviour in males and females. We show partial preventive effects of L. rhamnosus JB-1 supplementation. These results support the necessity to further investigate the potential negative long-term effects of early-life AB exposure, particularly with regard to neuropsychiatric disorders.

## Results

**Design of experiments.** BALB/c dams received low doses of penicillin V (AB group; 18:00–9:00, 31 mg kg$^{-1}$ per day) 1 week before pups' birth and up until weaning (postnatal day 21 (PND21)), so that the pups were initially colonized with an altered maternal microbiota and then received AB while nursing. Another group of dams (AB/JB1) received penicillin (18:00–9:00, 31 mg kg$^{-1}$ per day) and L. rhamnosus JB-1 (9:00–18:00, 10$^9$ c.f.u. per day). A control group (CT) received water and food ad libitum (Fig. 1).

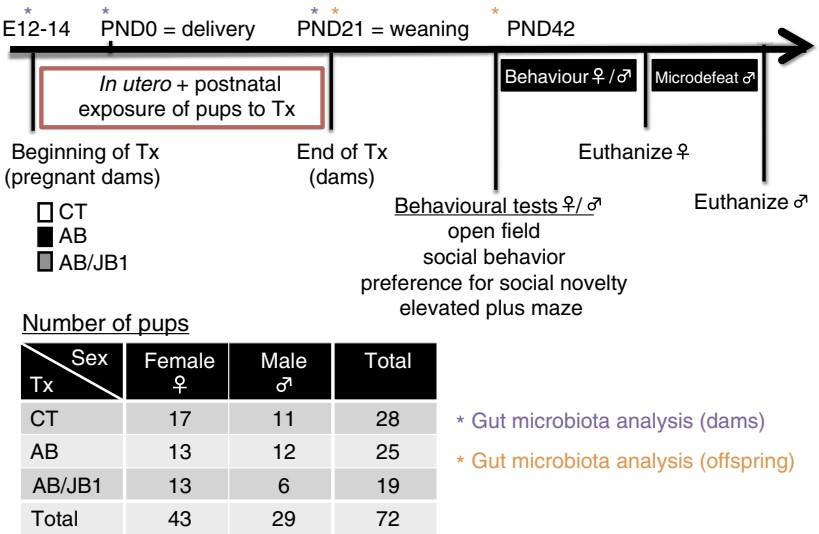

Number of pups

| Tx \ Sex | Female ♀ | Male ♂ | Total |
|---|---|---|---|
| CT | 17 | 11 | 28 |
| AB | 13 | 12 | 25 |
| AB/JB1 | 13 | 6 | 19 |
| Total | 43 | 29 | 72 |

\* Gut microbiota analysis (dams)

\* Gut microbiota analysis (offspring)

**Figure 1 | Study design.** One week before delivery (embryonic days 12–14), pregnant dams received either penicillin V (AB, $n = 4$) or penicillin V and Lactobacillus rhamnosus JB-1 (AB/JB1, $n = 3$) in drinking water. The control (CT, $n = 5$) group received only regular water. The treatment (Tx) was continued up until weaning (postnatal day 21, PND21). After weaning, the pups ($n = 72$) were separated from the dams and received regular water. At 6 weeks old (postnatal day 42, PND42), the offspring was subjected to a battery of behavioural tests. After the last test, females were euthanized while males were subjected to the microdefeat paradigm. Forty-eight hours following microdefeat, males were euthanized and blood, intestinal and brain tissues were collected.

**AB did not affect locomotor activity.** When the offspring (both sexes) were 6 weeks (postnatal day 42 (PND42)), they were first tested for locomotor activity. No main effect of treatment (two-way ANOVA: $F_{2,66} = 0.062$, $P = 0.940$), sex ($F_{1,66} = 0.542$, $P = 0.464$) or treatment × sex interaction ($F_{2,66} = 0.697$, $P = 0.502$) were observed on the total distance travelled in the open field, reflecting that general locomotor activity was similar in all mice (Fig. 2a)

Regarding time spent in the central area of the open field, a main effect of sex was found ($F_{1,65} = 4.98$, $P = 0.029$) while no effect of treatment ($F_{2,65} = 1.84$, $P = 0.17$) or treatment × sex interaction ($F_{2,65} = 1.69$, $P = 0.19$) were observed. The sex effect was due to the difference in time spent in the centre between males and females within the control group ($P = 0.005$ after Bonferroni correction; Supplementary Fig. 1).

**AB decreased anxiety-like behaviour in males.** Results of the elevated plus maze (EPM) showed no main effect of sex on the number of open arm entries (two-way ANOVA: $F_{1,65} = 0.014$, $P = 0.906$), however, a main effect of treatment ($F_{2,65} = 6.109$, $P = 0.004$) and treatment × sex interaction ($F_{2,65} = 4.15$, $P = 0.02$) were observed. Thus treatment affected anxiety-like behaviour and more importantly, both sexes were differently affected (Fig. 2b). In males, the number of entries in the open arms of AB-treated animals was higher than in CT ($P = 0.003$, *post hoc* test with Bonferroni correction) reflecting a decreased anxiety-like behaviour. In females, AB treatment did not alter anxiety-like behaviour (CT versus AB, $P = 0.325$), but mice treated with AB/JB1 exhibited a lower anxiety level compared to CT ($P < 0.001$) and AB groups ($P = 0.013$). There was no difference in anxiety-like behaviour between males and females within the CT group ($P = 0.994$). However, anxiety level was lower in males than in females within the AB group ($P = 0.021$) while there was a trend toward a lower anxiety-like behaviour in females compared to males in AB/JB1 group ($P = 0.08$). The same results were obtained for the time spent and the distance travelled in the open arms (Supplementary Fig. 2a,b). Anxiety-like behaviour is also assessed as a ratio of open to total arms entries[34]. No main effect of sex ($F_{1,65} = 0.68$, $P = 0.41$) but a main effect of treatment ($F_{2,65} = 7.11$, $P = 0.002$) and treatment × sex interaction ($F_{2,65} = 3.75$, $P = 0.03$) were observed (Supplementary Fig. 2c). Results of time and distance in the closed arms are presented in Supplementary Fig. 2d,e. The total distance travelled in the EPM, total number of entries as well as number of entries in the closed arms were similar in all groups reflecting no alteration of locomotor activity (Supplementary Fig. 2f–h).

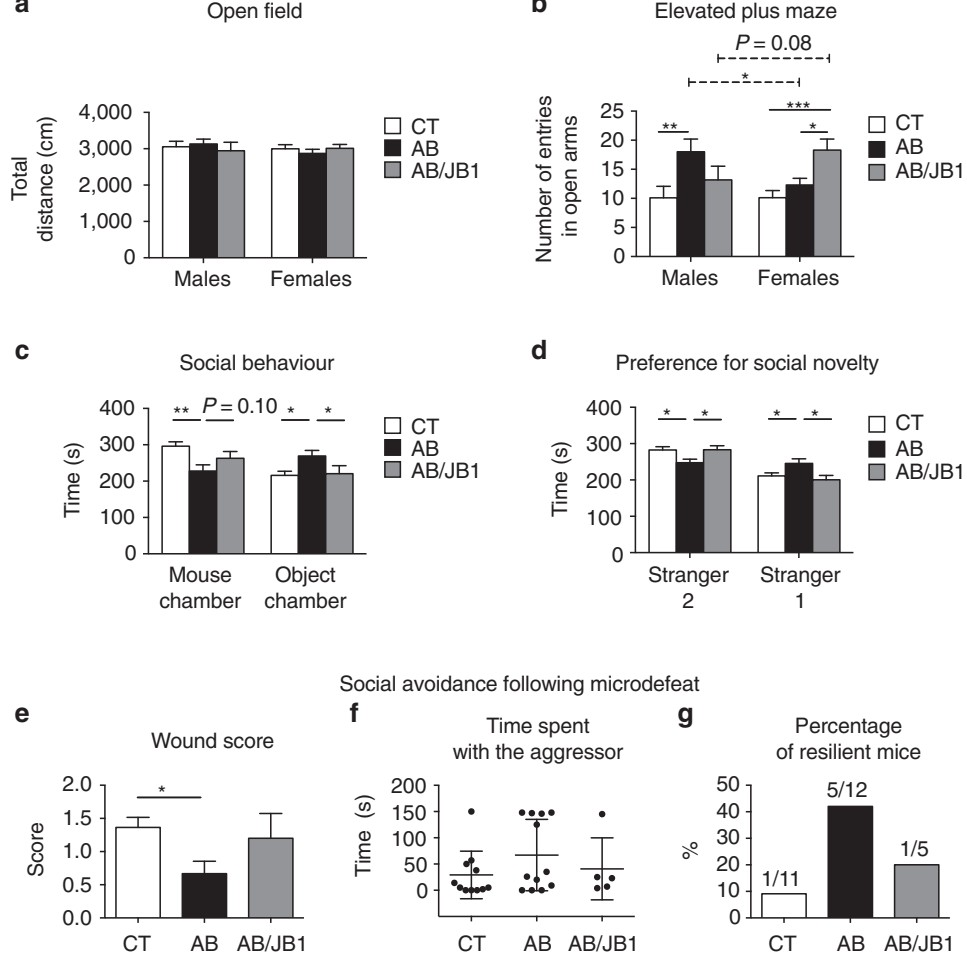

**Figure 2 | Effect of early life AB and AB/JB1 treatments on behaviour.** (**a**,**b**) Male and female mice were tested for locomotor activity in the open field and for anxiety-like behaviour by using the elevated plus maze (**a**; $n = 72$ (males, $n = 11$ CT, 12 AB, 6 AB/JB1: females, $n = 17$ CT, 13 AB, 13 AB/JB1), (**b**); $n = 71$ (males, $n = 10$ CT, 12 AB, 6 AB/JB1: females, $n = 17$ CT, 13 AB, 13 AB/JB1)) (two-way ANOVA). (**c**,**d**). Social behaviour and preference for social novelty were tested in the three-chambered apparatus ($n = 69$; males, $n = 11$ CT, 12 AB, 6 AB/JB1: females, $n = 16$ CT, 12 AB, 12 AB/JB1) (two-way ANOVA). (**e**–**g**) Males ($n = 11$ CT, 12 AB, 5 AB/JB1) were subjected to microdefeat and tested 24 h later for social avoidance. Results are means ± s.e.m. (except for graph F where results are means ± s.d.). *$P < 0.05$, **$P < 0.01$, ***$P < 0.001$. AB, antibiotic; AB/JB1, antibiotic and *L. rhamnosus* JB-1; CT, control.

These results suggest that early life AB treatment decreased anxiety-like behaviour in males as well as in females that also received JB-1.

**Lactobacillus prevented AB-induced decrease in sociability.** Mice were tested in the three-chamber apparatus for social interactions. Sociability is defined as the experimental mouse spending more time in the chamber containing the novel mouse than in that containing the novel object. Analysis revealed a main effect of treatment (two-way ANOVA: $F_{2,64} = 5.04$, $P = 0.009$) on the time spent in the novel mouse chamber but no main effect of sex ($F_{1,64} = 0.84$, $P = 0.363$) and no significant treatment × sex interaction ($F_{2,64} = 1.203$, $P = 0.307$). Male and female mice were not differently affected by the treatment. *Post hoc* tests showed that AB-treated mice spent less time in the novel mouse chamber compared to CT ($P = 0.002$) and AB/JB1 mice ($P = 0.10$; Fig. 2c). Also, there was a main effect of treatment ($F_{2,64} = 3.39$, $P = 0.04$) on the time spent in the chamber containing the novel object but no main effect of sex ($F_{1,64} = 0.005$, $P = 0.994$) or treatment × sex interaction ($F_{2,64} = 0.67$, $P = 0.517$). AB-treated mice spent more time in the object chamber compared to CT ($P = 0.019$) and AB/JB1 ($P = 0.049$) groups (Fig. 2c). These results show that early life AB treatment reduced social behaviour. However, concurrent ingestion of JB-1 prevented this reduction of sociability. Data on time spent in each chamber (centre, mouse, object) are presented for both sexes separately in Supplementary Fig. 3.

**Lactobacillus prevented AB-induced decrease in social novelty.** Mice were then tested for preference for social novelty by assessing time spent with a novel stranger or the initial stranger, respectively strangers 2 and 1. Two-way ANOVA revealed a main effect of treatment ($F_{2,64} = 3.92$, $P = 0.025/F_{2,64} = 3.82$, $P = 0.027$) but not sex ($F_{1,64} = 0.13$, $P = 0.72/F_{1,64} = 0.083$, $P = 0.78$) or treatment × sex interaction ($F_{2,64} = 2.32$, $P = 0.107/F_{2,64} = 1.60$, $P = 0.21$) on the time spent in the chambers containing either the stranger 2 or stranger 1, respectively. *Post hoc* tests showed that AB-treated mice spent less time with stranger 2 compared to CT ($P = 0.027$) and AB/JB1 groups ($P = 0.044$), and more time with stranger 1 compared with CT ($P = 0.049$) and AB/JB1 ($P = 0.019$; Fig. 2d). This shows that AB treatment decreased preference for social novelty, an effect that was prevented by concurrent supplementation with JB-1. When analysed separately for males and females, the preventive effect of JB-1 supplementation was more pronounced in females (Supplementary Fig. 4).

**AB increased aggression and reduced social avoidance.** BALB/c males were subjected to the microdefeat paradigm consisting of three sessions of 3 min during which they were physically stressed by an unfamiliar male CD-1 aggressor. All CT animals typically exhibited a submissive upright posture in response to CD-1 attacks (Supplementary Movie 1). Surprisingly, some AB-treated males fought back and exhibited upright offensive posture as well as rapid tail rattles characteristic of aggressive behaviour[35] (Supplementary Movie 2). Wound scores, measured at the end of the last defeat session, were significantly lower in the AB group compared to CT ($P = 0.048$, Kruskal–Wallis test; Fig. 2e). Twenty-four hours after the last defeat session, BALB/c males were tested for social avoidance with regard to a new, unfamiliar, CD-1 aggressor. Forty two percent (5/12) of AB-treated mice spent more time interacting with the aggressor and were therefore considered resilient to stress, compared to 9% (1/11) in CT and 20% (1/5) in AB/JB1 group (Fig. 2f,g). Mice treated with AB early in life had 3.5 times more risk to become resilient to stress compared to CT mice (relative risk = 3.5, CI 95% (0.56–22.08)).

**AB induced major gut microbial changes in dams and offspring.** Analysis of the gut microbiota was performed in dams before (T0) and after 1 and 4 weeks of treatment (T1, T2) and in the offspring at 3-week-old (PND21) and at 6-week-old (PND42). The pups were exposed to the treatment *in utero* (1 week before birth) and while nursing until weaning. At PND21, the treatment was stopped (Fig. 1).

To assess the effect of AB treatment on dams, we first analysed the beta diversity (between-sample diversity; Supplementary Fig. 5a (unweighted UniFrac) and Supplementary Fig. 6 (weighted UniFrac)). This revealed a strong AB effect where the AB and AB/JB1 groups were significantly different from CT ($P = 0.001$). When looking at the alpha diversity (within-sample diversity; Supplementary Fig. 7), there was no difference between groups at T0; at T1 and T2, the AB and AB/JB1 groups had lower diversity than CT but the results were not statistically different (t-tests) due to the small number of dams. The relative abundances of the two dominant phyla, Bacteroidetes and Firmicutes, were largely decreased after 1 week of AB treatment (Supplementary Fig. 5b). However, even if the treatment was continued for 3 more weeks, these two phyla returned to control levels at T2. The largest change was observed for the phylum Proteobacteria, which represented <0.5% of total operational taxonomic unit (OTU) counts in CT dams and around 80% in AB and AB/JB1-treated dams after 1 week of treatment. At T2, the level of Proteobacteria decreased significantly but remained higher than in the CT group. AB treatment also induced a decrease in Cyanobacteria and Actinobacteria (Supplementary Figs 5b and 8).

The microbiota of offspring of the AB and AB/JB1 groups was significantly less diverse than the control group (Supplementary Fig. 9 (alpha-diversity)) and also cluster separately from the control group ($P = 0.001$; Fig. 3a (beta-diversity unweighted UniFrac) and Supplementary Fig. 10 (weighted UniFrac)). Importantly, the dysbiosis remained until PND42, 3 weeks after ceasing treatment (Fig. 3). Analysis of relative abundances revealed that all bacterial phyla were changed following AB treatment (Fig. 3b–d). The relative abundances of two dominant phyla, Bacteroidetes and Firmicutes, were respectively decreased and increased in AB-treated mice. The most drastic change concerned the phylum Proteobacteria that was respectively 65- and 37 times more abundant in AB- and AB/JB1-treated mice compared to CT at PND21. Cessation of AB treatment after weaning allowed a significant reduction of Proteobacteria in AB and AB/JB1 mice at PND42 but its level was still largely higher than in CT. The phylum Deferribacteres was almost totally absent in all mice treated with AB and AB/JB1 while present in all CT mice, at both study time points. While the phylum Cyanobacteria remained stable over time in CT mice, the relative abundance of this phylum significantly increased from PND21 to PND42 in AB and AB/JB1 mice.

The significant decrease in Bacteroidetes following AB treatment was mainly due to a drastic reduction in the bacterial families S24-7, Prevotellaceae, Rikenellaceae and Odoribacteraceae. It is important to note that supplementation with JB-1 prevented the drop of S24-7 (Supplementary Figs 11 and 12). The increase in Firmicutes following AB treatment was due to a large increase in Lachnospiraceae, Clostridiaceae and Erysipelotrichaceae while the relative abundance of Lactobacillaceae was very low in AB and AB/JB1 groups (Supplementary Table 1). Supplementation with JB-1 prevented the changes in Lachnospiraceae and in Erysipelotrichaceae. A significant increase from PND21 to PND42 was observed for all treatment groups in

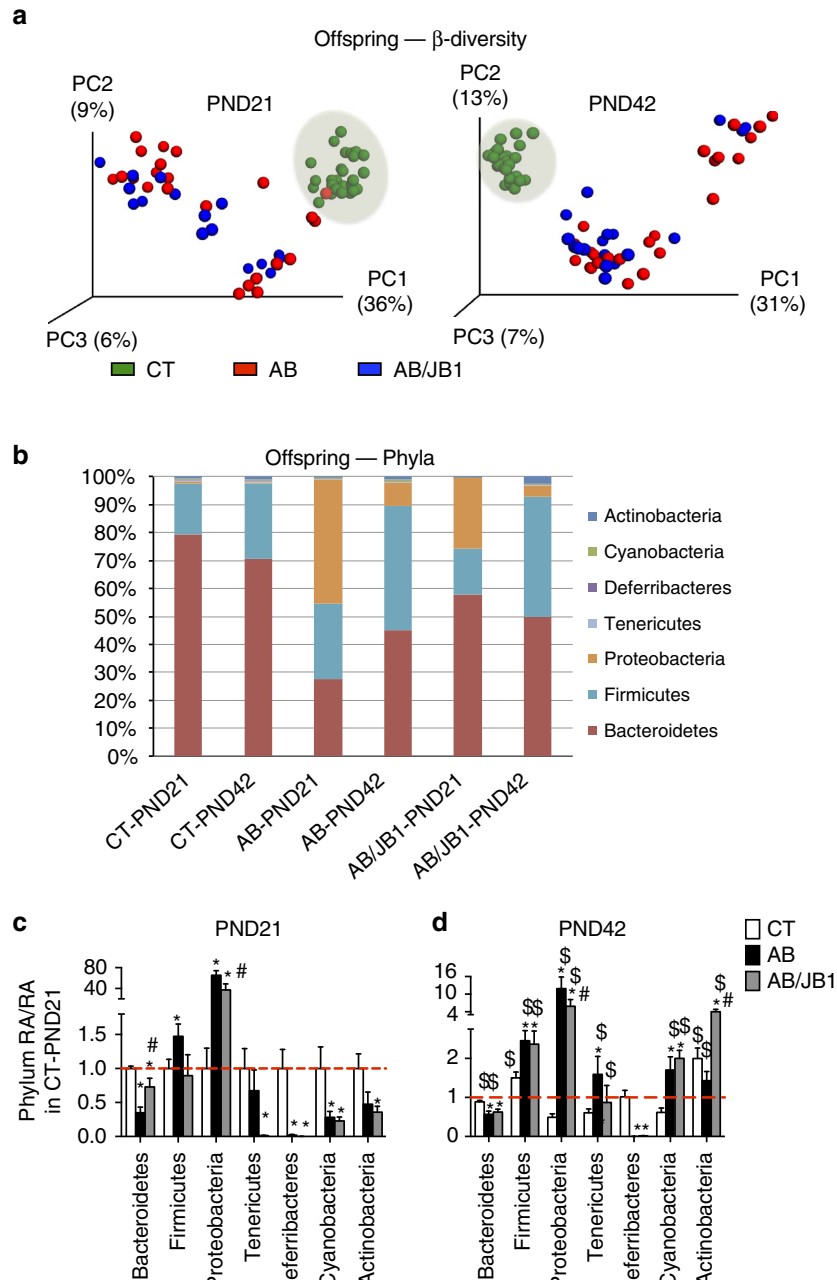

**Figure 3 | Gut microbiota composition in offspring at 3 and 6 weeks old.** (**a**) β-Diversity calculated with unweighted UniFrac matrix showing significant differences ($P = 0.01$) at both time points (PND21, 3 weeks old; PND42, 6 weeks old). (**b**) Relative abundance of bacteria phyla, expressed in percentage. (**c,d**) To better visualize the change induced by AB treatment in low-abundance phyla, the relative abundance of each phylum has been divided by the mean of relative abundance obtained in the CT group at PND21. The red line indicates the mean relative abundance of the phylum in the CT group at PND21, which corresponds to the value 1. Results are means ± s.e.m., $n = 68$ (26 CT, 24 AB, 18 AB/JB1) (mixed ANOVA). *$P < 0.05$ compared to CT group after Bonferroni adjustment for multiple comparisons (within the same study time point). #$P < 0.05$ compared to AB group after Bonferroni adjustment for multiple comparisons (within the same study time point). $P < 0.05$ compared to PND21, within the same experimental treatment group. AB, antibiotic; AB/JB1, antibiotic and *L. rhamnosus* JB-1; CT, control; PND, postnatal day; RA, relative abundance.

Ruminococcaceae and Dehalobacteriaceae (Supplementary Figs 11 and 13). The large increase in Proteobacteria was induced by the family Enterobacteriaceae, which represented 45% of all bacteria in AB-treated mice, 25% in AB/JB1 mice and <1% in CT mice (Supplementary Figs 11 and 14). Overall, male and female mice exhibited similar gut microbiota composition (Supplementary Fig. 15) except for the family Anaeroplasmataceae (Supplementary Fig. 16). To our knowledge, the role of Anaeroplasmataceae in host physiology remains unknown.

**AB did not induce change in gut barrier or inflammation**. Male and female mice treated with AB and AB/JB1 exhibited similar messenger RNA expression of cytokines (TNFα, IL-1β, IL-6 and IL-10) and chemokine (Cxcl15) and of tight junctions (TJs; occludin, zonula occludens 1 (ZO-1)) in colon compared to the CT groups (Fig. 4a–d). Similar results were obtained for the ileum (Supplementary Tables 2 and 3). The faecal albumin level was similar in all treatment groups in females, and lower in AB-treated males compared to CT (Fig. 4e,f). Altogether, these

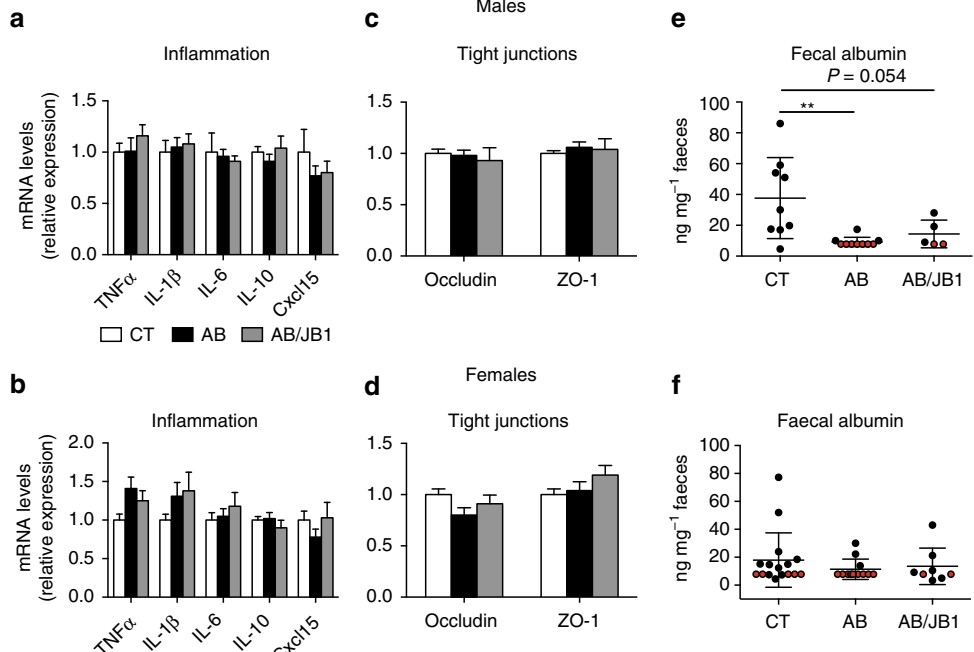

**Figure 4 | Assessment of inflammation and intestinal permeability in colon.** (**a**,**b**) Intestinal inflammation was evaluated by the mRNA expression of cytokines and chemokine. (**c**–**f**) Evaluation of intestinal barrier integrity was performed by measuring mRNA expression of tight junctions and fecal albumin concentration. Results are means ± s.e.m. (s.d. for **e**,**f**), $n = 28$ males (11 CT, 12 AB, 5 AB/JB1) and 43 females (17 CT, 13 AB, 13 AB/JB1) (one-way ANOVA). Red dot in graphs (**e**,**f**) indicates that the level of faecal albumin was below the detection limit. **\*\*$P < 0.01$. AB, antibiotic; AB/JB1, antibiotic and *L. rhamnosus* JB-1; CT, control.

data suggest that early life AB treatment did not disrupt the gut barrier and did not induce intestinal inflammation.

**AB induced changes in brain Avpr1b and cytokine expression.** We investigated whether changes in behaviour observed following early life AB treatment could be related to changes in gene expression in hippocampus and frontal cortex, two important brain areas involved in the regulation of behaviour. We found that the expression of arginine vasopressin receptor 1B (Avpr1b), known to be involved in aggressive behaviour[35], was substantially increased in the frontal cortex of both males and females treated with AB but not in the hippocampus (Fig. 5 and Supplementary Fig. 17).

BBB integrity was assessed by measurement of mRNA and protein expression of TJ occludin, and claudin-5 (Cldn5), key markers of the BBB integrity modulated by gut microbiota[18]. In the hippocampus, AB-treated males and females exhibited a significant increase in the mRNA expression of all TJ compared to CT mice (Fig. 6a,b). Western blot analysis confirmed the increase in TJ protein expression (Fig. 6e–g). AB/JB1 mice expressed hippocampal mRNA and TJ protein levels which were intermediate between CT and AB mice (Fig. 6a,b,e–g). In the frontal cortex, the mRNA expression of occludin was slightly increased in AB and AB/JB1 males while levels of cldn5 were not different from controls. In females, the levels of all TJ in the frontal cortex were similar in all groups (Fig. 6c,d). We conclude that early life AB treatment modified the BBB integrity in the hippocampus.

No changes in the expression of cytokines TNFα, IL-1β, IL-6, IL-10 and chemokine Cxcl15 (analogue of IL-8) were observed in the hippocampus of AB and AB/JB1 male or female mice (Fig. 6h,i). An increase in the expression of IL-6, IL-10 and Cxcl15 was seen in the frontal cortex of both sexes treated with AB (Fig. 6j,k). AB/JB1 males also expressed higher levels of IL-6, IL-10 and Cxcl15 in the frontal cortex compared to CT, while the

level of these markers in AB/JB1 females were intermediate between AB and CT mice (Fig. 6j,k). Interestingly, the expression of these cytokines was positively correlated with the expression of Avpr1b in both sexes (Supplementary Fig. 18), in line with the recently discovered role for vasopressin and its receptor as regulators of neuroinflammation[36].

These results show evidence of cytokine changes in the frontal cortex of both sexes induced by AB that were partially prevented in the AB/JB1 female group. They also suggest that the reinforcement of hippocampal BBB integrity may have prevented the increase in inflammatory markers, as suggested by negative correlations between TJ and IL-6, IL-10 and Cxcl15 (Supplementary Table 4). By contrast, in the frontal cortex, TJ were positively correlated with increased cytokines in both males and females (Supplementary Table 5).

**Brain cytokines were not related to systemic inflammation.** Systemic inflammation can lead to increased levels of cytokines in the brain and is associated with mood disorders[37]. Peripheral cytokines can reach the brain through a leaky BBB or through humoral, neural and cellular pathways[38]. We therefore assessed the levels of inflammatory cytokines in the systemic circulation to check if increases in brain cytokines could reflect a peripheral immune response. We found no increase in the serum levels of TNFα, IL-1β, IL-10 or in functional murine IL-8 homologues Cxcl1/KC, Cxcl2/MIP-2a and Cxcl5-6/LIX in AB-treated mice, suggesting that the increase in cytokines levels observed in frontal cortex of both sexes was not induced by a peripheral immune response (Supplementary Fig. 19).

**Discussion**
Beta-lactam AB are the most frequently prescribed drugs in infants and children[1] but their long-term effects on health have received scant attention until recently. Several clinical reports

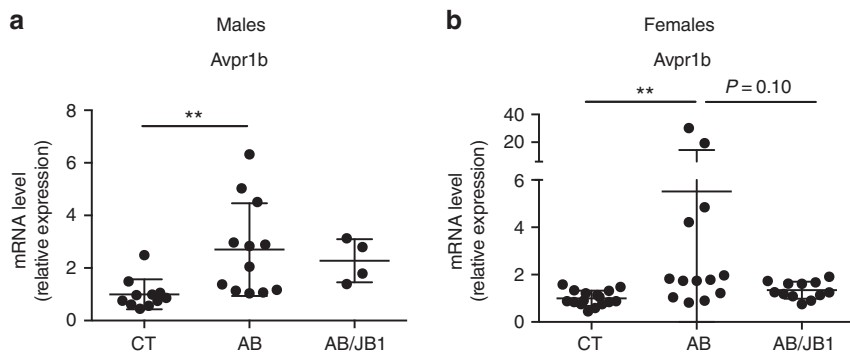

**Figure 5 | Brain expression of arginine vasopressin receptor 1B.** Avpr1b (arginine vasopressin receptor 1b) is known to be involved in the regulation of aggressive behaviour and brain cytokines changes. mRNA expression measured in the frontal cortex of (**a**) males ($n = 27$, 11 CT, 12 AB, 4 AB/JB1) and (**b**) females ($n = 42$, 17 CT, 13 AB, 12 AB/JB1; one-way ANOVA). Results are means ± s.d. **$P < 0.01$. AB, antibiotic; AB/JB1, antibiotic and *L. rhamnosus* JB-1; CT, control.

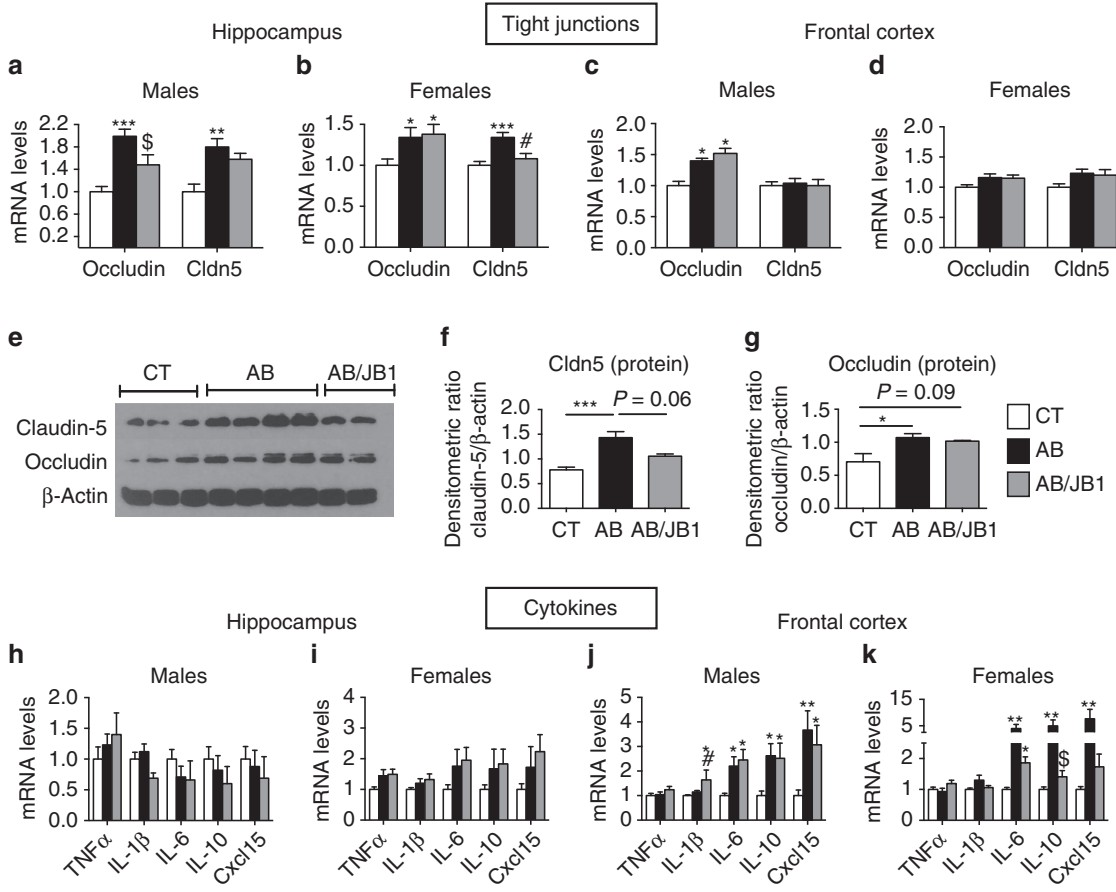

**Figure 6 | Brain tight junctions and cytokine expression.** (**a–d**) mRNA expression of tight junctions measured in the hippocampus and the frontal cortex of male and female mice. (**e–g**) Representative western blot of tight junctions proteins claudin-5 and occludin performed in hippocampus of male mice and quantification by densitometry normalized to the loading control β-actin (in total: $n = 5$ CT, 8 AB and 4 AB/JB1 mice). Full blots are shown in Supplementary Fig. 22. (**h–k**) mRNA expression of cytokines and chemokine in hippocampus and frontal cortex of male and female mice. Results are means ± s.e.m. (Hippocampus $n = 24$ males (9 CT, 11 AB, 4 AB/JB1) and 39 females (15 CT, 11 AB, 13 AB/JB1); frontal cortex $n = 28$ males (11 CT, 12 AB, 5 AB/JB1) and 42 females (17 CT, 13 AB, 12 AB/JB1)) (one-way ANOVA) *$P < 0.05$, **$P < 0.01$, ***$P < 0.001$ versus CT. #$P < 0.05$ versus AB and $P < 0.10$ versus AB. AB, antibiotic; AB/JB1, antibiotic and *L. rhamnosus* JB-1; CT, control.

have shown an association between early life AB use and an increased risk of developing allergies[3,4], inflammatory bowel diseases[7], obesity[8,9] as well as poorer neurocognitive outcomes later in life[10]. Altered gut microbial composition early in life may play a causal role in the lasting immune and metabolic changes associated with these diseases[12,13]. However, the impact of

intestinal dysbiosis induced by early-life AB exposure on brain function and behaviour remains unknown. Increasing evidence indicates that intestinal bacteria might communicate with the brain to induce changes in behaviour and neurochemistry[14–16]. While germ-free models or administration of high doses of AB in animals have revealed behavioural and cognitive alterations

associated with dysbiosis[22–26], they have unclear clinical relevance.

We tested first whether a low dose of penicillin (in our opinion, more clinically relevant than the higher AB doses used in some previous studies) early in life had long-term effects on gut microbiota composition, intestinal and BBB permeability, intestinal and brain cytokines expression and behaviour. Secondly, we investigated whether concurrent supplementation with *L. rhamnosus* JB-1, a psychoactive and neuroactive beneficial bacterium[31,32], can counteract the biological and behavioural effects of AB treatment. Since alteration of pre-weaning microbiota induced by continuous AB exposure leads to lasting immune and metabolic changes[11,12], penicillin V alone or concurrently with JB-1 were given continuously to pregnant dams 1 week before delivery and up until weaning. Ingested penicillin V is found in the fetal circulation and in breast milk[39]. The pups were therefore exposed to AB prenatally and postnatally, while nursing. We followed the design of a previous study that showed that sustained perinatal exposure to low-dose penicillin induced long-term changes in body weight and fat mass[12]. We found that early life AB treatment induced long-term changes in gut microbiota composition, BBB integrity and brain cytokines as well as behaviour. We also found that concurrent supplementation with *L. rhamnosus* JB-1 can attenuate certain deleterious effects of AB; however, this conclusion should be validated in further research, as the number of litters/dams, and the number of males in the AB/JB1 group, were relatively small. In addition, future studies should treat separate cohorts of mice with penicillin either *in utero* or from birth to weaning, to establish the importance of timing of AB exposure in the development of a specific adult phenotype. We also acknowledge that sustained exposure, through pregnancy and weaning, may not completely mimic the common use of AB in children. Future studies should also investigate the minimum number and duration of repeated AB exposures in early life that lead to long-term changes in host physiology.

We observed changes in the gut microbiota of AB-treated dams, especially at delivery of pups. Although we did not assess the dams' vaginal microbiota, we think that the pups were probably colonized at birth by an altered maternal microbiota because oral antibiotics have been shown to lower vaginal levels of *Lactobacillus* and increase the incidence of *E. coli* colonization in pregnant women[40,41]. Offspring gut microbiota was also largely altered by AB treatment with increased abundance of Firmicutes (Lachnospiraceae and Erysipelotrichaceae) and Proteobacteria, and decreased abundance of Bacteroidetes (S24-7, Prevotellaceae and Rikenellaceae) and Lactobacillaceae. In a recent mouse study[12], early life low-dose penicillin also reduced the abundance of Lactobacillaceae and Rikenellaceae which were considered potential protective candidates against long-lasting metabolic disturbances. In another mouse study[42], the increased relative abundance of Lachnospiraceae and decreased level of S24-7 observed at weaning were shown to predict diabetes and immune status later in life. We therefore hypothesize that some of these bacterial changes induced by early life AB could be involved in long-lasting behavioural alterations. Similar to dams, and as previously reported in other studies[25,43], we observed, in AB-treated mice, a large increase in Proteobacteria due to increased penicillin-resistant Enterobacteriaceae family[44], a potential source of lipopolysaccharides (endotoxins that can elicit strong immune responses in the host). Supplementation with *Lactobacillus* JB-1 partially counteracted the AB-induced increase in Enterobacteriaceae; we speculate that this may be due to JB-1 secreting short-chain fatty acids that affect the luminal pH[45] and thus creating a non-permissive environment for the growth of

Enterobactericeae. JB-1 supplementation also prevented the changes in S24-7, Lachnospiraceae and Erysipelotrichaceae. Nevertheless, the overall effects of JB-1 supplementation on penicillin-induced dysbiosis were modest.

Importantly, the drastic dysbiosis induced by AB observed at PND21 was still present at PND42, even after 3 weeks cessation of AB. However, it was not associated with changes in gut barrier integrity or gut inflammation. This is consistent with results obtained by Savage and Dubos[46] who observed no histological damage of intestinal mucosa following penicillin treatment.

Changes in microbial composition induced by early life AB treatment were accompanied by significant behavioural changes unlikely attributable to a direct toxic effect of penicillin on the brain since (1) penetration of penicillin into the CSF is very low in the absence of infection[47] and (2) the renal clearance of penicillin V is very rapid[47]. Further, AB treatment was stopped 3 weeks before beginning behavioural assessment. Future research could test potential direct effects of penicillin on the host by transferring faecal microbiota from AB-treated animals to germ-free mice and subsequent behavioural assessment; however, this procedure has several limitations[48]. The behavioural changes consisted mainly in decreased anxiety and social behaviour, reduced preference for social novelty and a surprising aggressive behaviour with preservation of general locomotor activity. Maternal care, that could play a role in shaping behaviour in adulthood, was not assessed in this current study. However, nests were carefully checked and no abnormalities or cannibalism occurred, suggesting that intestinal dysbiosis of dams did not affect maternal care, which is in line with a previous report by Sudo *et al.*[21] who did not find any effect of germ-free status on maternal behaviour.

While the gut microbial structure was similar between males and females, we found a sex effect regarding some behavioural data. A previous study, performed in germ-free mice, showed that CNS alterations occurred in a sex-specific manner and that reconstitution of a normal microbiota restored anxiety-like behaviour in males but not in females[49]. In our current study, anxiety-like behaviour was reduced in males treated with AB but not in females, while concurrent supplementation with JB-1 decreased anxiety in females but was associated with normal anxiety level in males. It has been previously shown that, in healthy adult BALB/c males, supplementation with JB-1 resulted in decreased anxiety levels and depression-like behaviour[32]. These JB-1-induced emotional changes were mediated by the vagus nerve, since vagotomized mice treated with JB-1 did not show such changes. No data are currently available on the effect of JB-1 on female mice with intact vagus nerves. Overall, our results highlight the fact that males and females do not always respond similarly to treatment and reinforce the importance of investigating sex differences in rodent studies. The mechanisms surrounding the sex differences in behaviour are not well understood. While oestrous cycle hormones could play a role, it is possible that other, yet unidentified, immunological or neuroendocrine factors or bacterial metabolites potentially influencing the vagus nerve could be at the origin of sex difference in the behavioural response to AB and JB-1 treatments. Faecal and blood metabolomic analysis as well as vagus nerve intervention could be tested in future studies to explore this possibility.

Reduced anxiety and social behaviour have been reported in germ-free mice[22,23] as well as in adult mice receiving a mixture of non-absorbable antimicrobials[24]. In the latter study, after a 2-week AB-washout period, the anxiolytic behaviour returned to normal. In our current study, even after 3 weeks of antibiotic cessation, mice still exhibited significant behavioural and social alterations suggesting that the pre-weaning bacterial colonization

is important to durably establish certain behaviours, as suggested previously[23].

Interestingly, we found that some of the AB-treated mice exhibited aggressive behaviour in the resident-intruder paradigm and spent more time interacting with the CD-1 aggressor. The microdefeat protocol has been validated for C57BL/6 males and, under control conditions, that is, when a naive C57BL/6 is subjected to microdefeat in the absence of other manipulation, it does not result in social avoidance behaviour[50]. To our knowledge, the microdefeat model has not been tested before in the more anxious BALB/c strain. However, a short version of chronic social defeat demonstrated that BALB/c spent less time interacting with the CD-1 aggressor mouse[51] compared to C57BL/6 mice. We found that, under control conditions, almost all untreated BALB/c were defeated and clearly avoided the CD-1 aggressor, while almost half of the AB-treated BALB/c were not socially defeated and indeed interacted with the aggressor. These aggressive AB-treated mice were characterized by a large increase in the expression of Avpr1b in the frontal cortex. Avpr1b is known to be involved in social and aggressive behaviours since deletion of this gene or administration of an Avpr1b antagonist to males promotes reduced aggressive behaviour[35,52]. Females were not tested for aggressive behaviour in our study for ethical reasons[53], but interestingly they also exhibited high levels of Avpr1b in the brain. Aggressive behaviour in AB-treated females needs to be explored in future studies.

The BBB begins to develop during intrauterine life and continues to mature during early postnatal stages[54]. The BBB protects against exposure to bacterial metabolites and potential toxins coming from the blood. TJ proteins that appear in the endothelium around embryonic days E11 to E13 (ref. 54) are mainly responsible for the barrier properties and are also functionally dependent on the presence of gut bacteria[30]. In our study, AB exposure of fetus started at E12–E14, a period which overlaps with the developmental window of TJ protein formation. Surprisingly and by contrast to germ-free mice, early-life AB treatment was associated with a reinforcement of BBB integrity as demonstrated by increased mRNA and protein expressions of occludin and claudin-5 in the hippocampus, but not in the frontal cortex. Since inflammatory cytokines are known to induce behavioural and cognitive impairments, we speculate that the reinforcement of BBB integrity may have prevented the production of inflammatory cytokines in the hippocampus, and that might be a protective mechanism to preserve hippocampal-dependent tasks like learning and memory. By contrast, AB treatment induced a large increase in specific cytokines and chemokine in the frontal cortex of both sexes. Previous studies[55–57] have shown that chronic stress induces anxiety-like behaviour that was associated with increase in Ly6C[hi] monocytes trafficking to the frontal cortex, increased brain expression of IL-1β and TNFα, and reduced IL-10, suggesting a pro-inflammatory state. However, Möhle et al.[58] recently reported that antibiotic treatment decreased the number of brain Ly6C[hi] monocytes, which was associated with decreased neurogenesis. Supplementation with probiotics and physical exercise were able to restore hippocampal neurogenesis by increasing the number of Ly6C[hi] monocytes in the brain. Consequently, increased influx of brain Ly6C[hi] monocytes, an important cell population serving as mediator between the periphery and the brain, was considered to be a beneficial response to ensure neurogenesis. Other studies have also emphasized the neuroprotective functions of bone marrow-derived microglia in Alzheimer's disease[59] and in other brain diseases where infiltrating monocytes-derived macrophages could become resolving cells and secrete anti-inflammatory cytokines[60]. We did not assess the number of Ly6C[hi] monocytes in our study but cytokines IL-6, IL-10 and chemokine Cxcl15 (IL-8) were significantly increased in the frontal cortex of mice treated with AB, whereas TNFα and IL-1β, the two main markers of the pro-inflammatory response, were not changed. IL-6 has a dichotomous action in the brain with both pro- and anti-inflammatory function helping to maintain BBB integrity[61,62]. We found strong positive correlations between occludin and IL-6 in the frontal cortex that could suggest a beneficial effect of IL-6 in preserving BBB integrity. IL-10 is a typical anti-inflammatory cytokine. Intracerebroventricular administration of IL-10 almost completely inhibits LPS-induced brain TNFα and IL-1β production while IL-6 expression is not affected[63]. In our study, it is plausible that IL-10 induction promoted a negative feedback on pro-inflammatory TNFα and IL-1β in the frontal cortex. The role of Cxcl15 in the brain has received little attention but has been shown to stimulate the production of neurotrophic factors[64]. We speculate that the change in cytokines expression that we found in the frontal cortex of AB-treated mice does not represent a typical pro-inflammatory status but could be a specific response of the brain to cope with early life AB treatment. Furthermore, immune examination of serum, ileum and colon from treated mice revealed no evidence of any systemic or peripheral inflammatory changes suggesting that brain cytokines upregulation induced by AB is a localized response. Finally, vasopressin and its receptor Avpr1b have been shown to exacerbate the production of brain inflammatory cytokines and chemokines[36,65]. In our study, we found correlations between Avpr1b and IL-6, IL-10 and Cxcl15 in the frontal cortex of both males and females, but not the hippocampus, suggesting that Avpr1b, in addition to being involved in social and aggressive behaviours, could also be involved in the regulation of brain cytokines expression.

While all these data obtained in rodents cannot be directly extrapolated to humans, they add support to the necessity to carefully consider the potential negative long-term effects of early-life AB exposure. The lasting dysbiosis and the persistence of cytokine change in the frontal cortex associated with aggression and reduced social interactions and anxiety raise questions regarding the important role of this brain area in the development of autism and other neuropsychiatric disorders. The partial preventive effects of a Lactobacillus given concurrently with the AB early in life are intriguing and warrant further investigation of their potential to attenuate some of these possibly noxious long-term effects.

## Methods

**Study design.** Male and female BALB/c mice (breeding pairs), 6–8 weeks old, were acquired from Charles River (Montreal, QC, Canada) and allowed to acclimatize in the housing facility for at least 1 week. Breeding of mice was organized as follows: a single female mouse was placed in the male's cage for 48 h. Pregnancy was confirmed by increased weight (>3 g within 8 days following mating). One week before delivery, pregnant females were housed singly with nesting material and treatment (Tx) was started (number of days of Tx before delivery = 7.3 ± 1.1). Pregnant females were treated with either drinking water (control group defined as CT, $n = 5$) or penicillin V (antibiotic group defined as AB, $n = 4$) or penicillin V and *Lactobacillus rhamnosus* JB-1 (defined as AB/JB1, $n = 3$) until weaning of pups (postnatal day 21). Pups were therefore exposed to the treatment prenatally and during the early postnatal life. This period is referred to as 'early life' in the description of the results. Since the dams were the animals treated and not the pups, it was impossible to randomly assign the pups to a treatment group. At weaning, male and female offspring were separated from the dams and housed 3–5 per cage and received regular drinking water and standard rodent chow *ad libitum*. A battery of behavioural tests was started when the offspring reached 6-weeks old (postnatal day 42), with 2 days of rest between each test. The microdefeat was performed at the end of the experiment, only in males. Mice were killed by decapitation the day following the last test in order to collect trunk blood and tissue samples. A total of 72 mice were used in this study (Fig. 1). Sample size of this exploratory study could not be calculated here as the effect size was unknown. For the microdefeat paradigm, male CD-1 retired breeders were obtained from Charles River. As per our ethical approval, only male mice were used in the social defeat experiments. All animals were housed under 12 h light/12 h

dark cycle. All experiments followed the guidelines of the Canadian Council on Animal Care and were approved by the McMaster Animal Research Ethics Board.

**Treatment with antibiotic and *L. rhamnosus* JB-1.** Dams received AB and *L. rhamnosus* JB-1 treatments through the drinking water in order to avoid stress induced by oral gavage. AB treatment consisted in penicillin V (Sigma-Aldrich, MO, USA) that is absorbable by the gastro-intestinal tract[66] and found in breast milk[39]. Penicillin V crosses placenta and passes into fetal circulation[47]. Dams received a low dose of penicillin V that, in our opinion, is clinically relevant (50,000 U kg$^{-1}$ per day or 31 mg kg$^{-1}$ per day)[47,67]. Since water consumption increases significantly during the gestational and lactating periods and is influenced by the number of pups per litter, the dose of penicillin V was adjusted to body weight and water consumption of dams, measured twice weekly. *L. rhamnosus* JB-1 was obtained as a gift from Alimentary Health, Cork, Ireland and is the same bacterial strain we have used previously[32]. It was also administered in drinking water (10$^9$ c.f.u. per day) in which it remains viable for more than 12 h. Dams from the AB group received regular water during the day (9:00 to 18:00) and penicillin V during the night (18:00 to 9:00). Dams from AB/JB1 group received *L. rhamnosus* JB-1 during the day and penicillin V during the night. This schedule, reported in a previous study[27], has been used to avoid the adverse effect of AB on the bacteria. We found using quantitative PCR that *L. rhamnosus* was detectable in faeces of all dams before treatment and only in dams of the AB/JB1 group after treatment (Supplementary Fig. 20), proving that the protocol using two separated drinking bottles alternately, is efficient in maintaining the presence of some *L. rhamnosus* in the gut. All bottles of water containing bacteria were shaken three times per day.

**Behavioural testing.** All behavioural tests were recorded by a video camera or connected to a computer and data analysis was performed by using Motor Monitor software (Kinder Scientific, Poway, CA) or EthoVision XT software (Noldus, Leesburg, VA). The experimenter was not blinded to the group allocation when assessing behaviour.

*Open field.* Mice were tested during the dark phase under dim-light conditions. After a 1 h period of habituation in the testing room, animals were placed in the enclosure for 30 min. Distance moved in the open field and rearing were recorded using Motor Monitor software. The apparatus was thoroughly cleaned with water and dried between each animal.

*Three-chamber sociability test.* Mice were tested in the three-chamber apparatus during the light phase, following a 30 min period of habituation in the testing room. The apparatus consists of three rectangular Plexiglas chambers whose dividing walls possess small openings that allow access into each chamber. The experimental mouse was first placed in the centre chamber while the doorways were closed and allowed to explore for a 5-min habituation period. In the second phase of the test, an unfamiliar mouse (strain- and sex-matched) was placed within an inverted wire cup in one of the outer chambers while an empty wire cup was placed in the other outer chamber. The doors to the outer chambers were opened and the sociability trial was conducted for 10 min during which the experimental mouse was allowed to explore the three chambers. Time spent in each chamber was recorded by a video camera positioned over the apparatus and analysed by using EthoVision XT software. Sociability is defined as the experimental mouse spending more time in the chamber containing the novel mouse than in the chamber containing the novel object. In the third phase of the test, a second novel mouse (stranger 2) is placed inside the previously empty wire cup while the initial novel mouse (stranger 1) remains inside its cup. The experimental mouse is given 10 min to explore all the three chambers. Preference for social novelty is defined as more time spent in the chamber with stranger 2 than time spent in the chamber with stranger 1. The apparatus was thoroughly cleaned with water and dried between each animal.

*Elevated plus maze.* Mice were tested in the EPM apparatus that is elevated at 76 cm off the ground, consists of four arms—two open arms and two closed arms made with black Plexiglas walls. Mice were transported to the behavioural testing room for a 30 min habituation period. The mouse was then placed in the intersection of the four arms, facing open arm, and allowed to explore for 5 min. The EPM was connected to a computer and behavioural data were analysed by using Motor Monitor software. Time spent, distance travelled and number of entries in the open arms were used to assess anxiety-like behaviour. The apparatus was thoroughly cleaned with water and dried between each animal.

*Microdefeat stress and social avoidance test.* The microdefeat paradigm is a short (acute) version adapted from the chronic social defeat described by Krishnan *et al.*[68] and by Golden *et al.*[50]. The microdefeat is a model that was developed initially to measure increased susceptibility to stress. It can reveal susceptible phenotype if an animal's stress threshold is shifted by the experimental manipulation. The microdefeat has been described in C57BL/6 male mice only and under control condition, this protocol does not induce social avoidance. There are currently no widely accepted versions of microdefeat in females. In our study, microdefeat has been performed in male BALB/c as follows. First, CD1 aggressors (retired breeders >4–5 months old) were screened for consistent attack latencies (<60 s on three consecutive screening sessions with BALB/c intruders called screeners). CD-1 mice were singly housed throughout the experiment. Successful application of social defeat stress is dependent on appropriate selection of CD-1 mice with consistent levels of aggressive behaviour. Within the same day, BALB/c

mice are forced to intrude into the space territorialized by the larger and aggressive CD-1 mouse for three sessions of 3 min, followed by a rest period 15 min in the home cage between each exposure. This situation leads to intruder aggression and subordination. After the last resident–intruder session, BALB/c mice are housed singly (with free access of food and water) and a wound score (0: absent, 1: light, 2: moderate, 3: severe) was assigned. BALB/c mice were tested for social avoidance 24 h later. Social avoidance test was performed in a rectangular Plexiglass box, where an unfamiliar CD-1 aggressor was placed under a wire cage at one end of the arena. The box was virtually split into two parts, an interaction zone (close to the wire cage) and a non-interaction zone at the other end of the arena. The BALB/c mouse was allowed to explore the arena for a first trial of 150 s when the aggressor was absent (empty wire cage) and then for a second period of 150 s when the aggressor was present. A social interaction ratio was calculated as follows: 100 × (time spent in the interaction zone when the aggressor is present/time spent in the interaction zone when the aggressor is absent). Mice with an interaction ratio <100 were considered susceptible to stress while mice with an interaction ratio >100 were resilient to stress. Results were analysed by using EthoVision XT software.

**RNA extraction and RT–qPCR analysis.** After decapitation of animals, gut (distal ileum and colon) and brain (frontal cortex and hippocampus) were quickly dissected and put into RNA*later* solution (Ambion, Life Technologies, CA, USA). Tissues were incubated overnight at 4 °C then transferred at − 20 °C until further processed. Following tissue homogenization, total RNA was extracted using TRIzol Reagent (Ambion, Life Technologies). One microgram RNA was then converted into cDNA using SuperscriptIII First-Strand Synthesis Supermix (Invitrogen, CA, USA). Diluted or non-diluted cDNA was used as template for qPCR reaction using PowerUp SYBR Green Master Mix (Applied Biosystem, Life Techologies, TX, USA) containing ROX dye Passive Reference. The qPCR reactions were performed in the fast mode (UDG activation 50 °C, 2 min; Dual-Lock DNA polymerase 95 °C, 2 min; denaturation: 95 °C, 1 s; annealing/extension 60 °C, 30 s; number of cycles: 40–50) by using QuanStudio3 machine (Applied Biosystem). Data were normalized to the endogenous control GAPDH and the relative quantification was analysed using the ΔΔCt method. The experimental treatments did not affect GAPDH expression in any of the tested tissues (Supplementary Fig. 21). Primers were designed with Primer Express Software and used at a concentration of 300 nM. Primer sequences are listed in Supplementary Table 6.

**Western blotting.** Brain tissues (hippocampus and frontal cortex) were quickly dissected following decapitation and stored at − 80 °C. Samples were homogenized and lysed (45 min on ice) in protein lysis buffer (100 μl per 10 mg; 0.05 M Tris-HCl, 0.15 M Nacl, pH 7.4) containing 1% Triton X-100, 1 mM EDTA, 10 mM NaF, 0.5% sodium deoxycholate, 0.1% SDS, 1 mM PMSF, 1 mM Na$_3$VO$_4$ and protease inhibitor cocktail (1 tablet per 10 ml, cOmplete Roche, Sigma-Aldrich) and centrifuged at 15,000$g$ for 10 min at 4 °C. The supernatant was collected and protein concentration was measured by the Lowry method (DC Protein Assay, Biorad). Proteins (20 μg) were fractionated on a 12% SDS–polyacrylamide gel electrophoresis gel and transferred to 0.2 μm polyvinylidene difluoride membranes (GE Healthcare Life Sciences, Germany). Membranes were blocked in 0.1% Tween-TBS 20 with 5% nonfat dry milk for 1 h at room temperature, then probed with rabbit antibodies to occludin (Invitrogen 40–4700, 1:1,000, overnight, 4 °C), claudin-5 (Invitrogen 34-1600, 1:1,000, overnight, 4 °C) or β-actin (Bioss bs-0061R, 1:5,000, 2 h, room temperature). After extensive washing, membranes were incubated for 1 h at room temperature with a goat anti-rabbit IgG-peroxidase secondary antibody (Invitrogen A0454, 1:20,000), then visualized with Amersham ECL western blotting detection reagents (GE Healthcare Life Sciences) and developed in the dark room. Densitometry was quantified with Image J[69].

**Assessment of intestinal barrier permeability.** Evaluation of intestinal barrier integrity was performed by measuring the mRNA expression of tight junction proteins occludin and zonula-occludens 1 (ZO-1), two key markers of paracellular permeability[70], in ileum and colon and by the faecal albumin concentration. The latter is a good indicator of disrupted intestinal barrier function and has been shown to be comparable and correlate well with the orally administered FITC-dextran method[71]. Faecal albumin was determined by ELISA (Bethyl Labs, TX, USA) following manufacturer's guidelines.

**Serum cytokine assays.** Trunk blood was collected into sterile tubes, allowed to clot at room temperature, spun down for 10 min at 850$g$ and serum was stored at − 80 °C. Cytokines were analysed by using the mouse cytokine 32-plex discovery assay (Eve technologies, AB, Canada).

**16S rRNA gene sequence analysis.** Stool samples were collected in sterile tubes and stored at − 80 °C until further analysis. DNA from 176 stool samples was extracted using the PowerSoil HTP DNA Isolation Kit (MoBio, USA) according to the manufacturer's instructions with a beadbeater (BioSpec, USA) set on high for 2 min. Following DNA extraction, the V4 variable region of the bacterial 16S rRNA gene

was amplified by PCR using the 515F and 806R (each well received a separate 515F barcoded primer). 515F- (barcode) 5′-AATGATACGGCGACCACCGAGATCTAC ACGCTAGCCTTCGTCGCTATGGTAATTGTGTGYCAGCMGCCGCGGTAA-3′ and 806R 5′-CAAGCAGAAGACGGCATACGAGATAGTCAGTCAGCCGGAC TACHVGGGTWTCTAAT-3′ (ref. 72). PCR reactions were carried out with the Primestar taq polymerase (Takara, Japan) for 30 cycles of denaturation (95 °C), annealing (55 °C) and extension (72 °C), and a final elongation at 72 °C. Products were purified using AMPure magnetic beads (Beckman Coulter, USA) and quantified using Pico-green dsDNA quantitation kit (Invitrogen, USA). Samples were then pooled at equal concentrations (50 ng μl$^{-1}$), loaded on 2% E-Gel (Thermo Fisher, USA) and purified using NucleoSpin Gel and PCR Clean-up (Macherey-Nagel, Germany). Purified products were sequenced using the Illumina MiSeq platform (Genomic Center, Faculty of Medicine, BIU, Israel). Data analysis was performed using the Quantitative Insights into Microbial Ecology (QIIME) pipeline version 1.8.0 (refs 72,73). Paired-end sequences were joined using fastq-join, demultiplexed and quality filtered with an average quality threshold of 25. Chimeric sequences were identified using USEARCH and removed, and reads were clustered into OTUs using the open reference UCLUST method against the GreenGenes 08/13 database[74], with a cutoff of 97% sequence identity. Core OTUs were calculated by filtering for OTUs presents in at least 50% of subjects in the same treatment group. Analyses were performed on the core OTUs using a rarefied table of 10,200 sequences per sample. In addition, alpha diversity was estimated using Faith's phylogenetic diversity (PD whole tree) and beta diversity was calculated using weighted and unweighted UniFrac[75]. Significance between groups for distances was assessed using $t$-tests as implemented in 'make_distances_boxplots.py' script in QIIME 1.8.0.

**Statistical analysis.** Results of behavioural tests were analysed by a two-way ANOVA with sex (male versus female) and treatment (CT, AB, AB/JB1) as between factors. Homogeneity of variance was tested with Levene's test. When the sex × treatment interaction was significant, *post hoc* tests using Bonferroni correction were performed. Analysis of the bacterial relative abundance in dams and offspring was performed by using mixed ANOVA with time (T0, T1, T2 or PND21, PND42) as a within-factor (repeated measure) and treatment (CT, AB, AB/JB1) as a between-factor. For the dams, assumption of sphericity was checked with Mauchly's test and Greenhouse–Geisser correction was applied to produce a valid F-ratio when condition of sphericity was not met. For the offspring, homogeneity of variances for each combination of the groups (time and treatment) was checked with Levene's test. If the time × treatment interaction was significant, *post hoc* tests were performed with Bonferonni correction (when homogeneity of variances was met) or with Games–Howell procedure (when homogeneity of variances was not assumed). Significant outliers were identified graphically (with boxplot) and by using the 'studentized residual'. Since males have been subjected to the social defeat while females have not been stressed, males and females were analysed separately regarding the biological assays (serum, gut and brain samples) by using one-way ANOVA (or Kruskal–Wallis test if assumptions of normality and homoscedasticity were not met). Tukey *post hoc* tests were performed for pairwise comparisons. Results were considered statistically significant when $P < 0.05$. All tests were two-tailed. Statistics were performed with SPSS 23.0.

**Data availability.** The 16S rRNA gene sequence data have been deposited in the European Bioinformatics Institute (EBI) database with accession code ERP021539. The authors declare that all other relevant data supporting the findings of this study are available within the paper and its Supplementary Information files, or from the corresponding authors on request.

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

## Acknowledgements

We thank Dr Jane Foster for teaching brain dissection, Dr Karen-Anne McVey Neufeld for performing biological assay, Dr Mike Surette for his help in DNA extraction and Harman Bathiati for her help collecting tissues during the killing of mice. We also kindly thank Dr Sébastien Matamoros for his precious help in bioinformatics, and gratefully acknowledge Alimentary Health Inc, Cork, Ireland for their generous gift of *Lactobacillus rhamnosus* JB-1. We acknowledge grant support from the US Office for Naval Research (ONR) (N00014-14-1-0787). S.L. is a recipient of a post-doctoral fellowship from the ONR and received funds from FSR (Fonds Spécial de la Recherche), Belgium. P.F. and O.K. are supported by the Canadian-Israel Health Initiative, jointly funded by the Canadian Institutes of Health Research, the Israel Science Foundation, the International Development Research Centre, Canada and the Azrieli Foundation. L.B.B. is a Post-doctoral Researcher from the F.R.S.-FNRS (Fond National de la Recherche Scientifique, Belgium).

## Author contributions

S.L., P.F. and J.B. conceived and designed the project. S.L. treated mice and performed behavioural testing, RT–qPCR, western blot and statistical analysis. F.M.M. and A.M.S. were involved in animal experiments and performed the faecal albumin assay. L.B.B. performed experiments on *L. rhamnosus*. O.K. and H.B.-A. performed gut microbiota sequencing and with S.L. and E.C. analysed the microbiome data. S.L., P.F. and J.B. analysed the data and wrote the manuscript.

## Additional information

**Competing interests:** The authors declare no competing financial interests.

