## [Peer Review File · Nature Communications]

Reviewers' comments:

Reviewer #1 (Remarks to the Author):

Excessive and inappropriate antibiotic use in early life is a growing concern and a growing number of preclinical studies now indicate that this might have implications for CNS function as well. In addition to more restraint in the unnecessary use of broad-spectrum antibiotics, effective intervention strategies to counteract their detrimental impact are urgently required. The current report investigated the impact of chronic low-dose of penicillin in early life on behavioral and CNS abnormalities in the offspring during adulthood. In addition, the ability of concurrent supplementation with *Lactobacillus rhamnosus* JB-1 to prevent the emergence of these alterations was assessed. The main finding of the study is that the antibiotic treatment, which induced sustained changes in the gut microbiota, decreased anxiety, sociability and social avoidance, and increased aggressive behaviour. Interestingly, some of these alterations were sex-specific but were not accompanied by alterations in gut barrier function or systemic inflammation despite evidence of neuroinflammation in the frontal cortex of both male and female animals.

The manuscript offers a number of important insights and some very intriguing results with the differential effects in male and female animals particularly notable. The scope of the assessments is also impressive. Nevertheless I do have a number of concerns that require input for the authors prior to further consideration.

(1) It would seem that the study design actually includes a number of insults to the developmental trajectory of the early life microbiota. As the authors note, the maternal microbiome is also altered and there is likely differential postnatal microbiota transmission as well as antibiotic effects. This deviates quite a bit from the study by Cox et al, cited as inspiration for this protocol, in which the low dose penicillin was delivered from birth. I am curious also as to whether the different treatments impacted on maternal care? Was the dams physiology altered as well as their microbiota and could this have impacted in utero? The design also does not permit analysis of the impact of the lactobacillus strain on it's own, something I don't believe has been assessed previously in a similar context.

(2) The authors consider the dose of antibiotic more clinically relevant. This might be the case but this equivalence is somewhat lost in the chronic sustained treatment. The introduction and discussion should really note this important caveat.

(3) The question being asked here is clearly important and justified but the introduction meanders a bit and would benefit from a more streamlined revision.

(4) Despite some of the reservations noted in (1) above, the intervention produces robust alterations at the behavioral and microbiota level. However, I find the biological analyses a little difficult to interpret since as far as I understand, there is no assessment in stress-naïve male animals. Although the authors do point out that the male and female animals are analysed separately because of this, the results in the male animals may well be a result of a stress of social defeat X antibiotic interaction. Either alone are known to alter the microbiota as well as many of the biological features reported (e.g. BBB, intestinal permeability). Was the microbiota assessed at the end of the behavioural testing?

(5) The manuscript states that mice were sacrificed by decapitation the day following the last test in order to collect trunk blood and tissue samples. Does this mean that the females (who were not assessed in the social defeat paradigm) were sacrificed at a different time point to the male animals? Or that they had a greater recovery time following the last test if sacrificed at a time point equivalent to the male animals?

(6) The authors report that although there was no treatment effect on tight junction protein

expression in the frontal cortex, they were positively correlated with neuroinflammation in both males and females. I am not convinced of the relevance of this finding – perhaps a revised manuscript could elaborate on this point since I'm not sure how an intact BBB could be linked to the observed inflammatory alterations?

(7) It is stated that qPCR was used to show that *L. rhamnosus* was detectable in feces of dams that received both AB and JB-1. I would argue that it would be more important to determine whether AB treatment affected the abundance of JB-1 compared to JB-1 treatment alone. I note as well that the protocol cited by the authors (Wang et al.) in support of limiting the adverse effect of penicillin on JB-1 was conducted and validated using a different strain and may not generalise.

(8) Any revised submission should include measures of alpha diversity of the microbiota.

(9) For the offspring, there were only 6 male animals on the combined JB-1/AB intervention. The possibility that this resulted in an underpowered analysis should be noted as a limitation in the discussion.

(10) Could the authors comment in the revised submission on the value of a microbiota transfer to resolve the question of whether it is the altered microbiota or the antibiotics that induced the observed alterations?

(11) Am I correct in thinking that the microbiota analyses reported for the offspring in the supplementary data does not include a breakdown according to sex?

(12) The discussion would benefit from a more detailed consideration of the sex-specific behavioral alterations in contrast to the similarities for the neuroinflammation data.

Reviewer #2 (Remarks to the Author):

Title: Early life exposure to clinical dose penicillin induces long-term changes in gut microbiota, brain, inflammation and behavior

This was a study to determine the effects of penicillin in late pregnancy through weaning on gut immunity, brain, and behavior. The data indicate that the penicillin changed the microbiome in pregnant dams and offspring, and also resulted in inflammation in the frontal cortex and differences in gene expression for arginine vasopressin receptor. The penicillin also reduced anxiety in the offspring. Many, but not all, of these effects were prevented by the administration of *L. rhamnosus* JB-1. Understanding the impact of early life antibiotics on infant development is important, and with the growing evidence that the microbiota contribute to the gut-brain axis, this is a study of importance to the field. The data are interesting, but there are some comments that the authors should address.

Major Comments:

1. There seems to be a control group missing (namely no antibiotic, but with JB1). Please justify why a fully crossed 2 x 2 experimental design (control vs. antibiotic x vehicle vs. JB1) was not used.

2. The sample size is quite low. Although there is a total of 72 pups (which is a good sample size), only 5 dams were in the control condition, 4 dams were antibiotic treated, and 3 dams were given probiotic. There is some suggestions in the literature that sample sizes should really be based on

the # of litters (not # of offspring).

3. It is not entirely clear what the real purpose/conclusion of the study is. As it is currently written, the manuscript is really just a description of the effects of prenatal/early life antibiotics on mouse biology without placing the results into context. For example, the first paragraph of the discussion indicates that penicillin AB are frequently prescribed in early life, but their effects on adulthood diseases such as allergies, IBD, and obesity are not known. Why does this study focus on the brain and behavior and not the aforementioned diseases? Is there evidence that early life antibiotics lead to mental illness later in life?

4. Related to the point above, how do the authors interpret the increased neuroinflammation that has been shown by multiple groups to lead to increased anxiety in mouse models? Why do the animals in this study have reduced anxiety, but increased neuroinflammation?

Specific Comments:

1. Penicillin was given in the drinking water. Mice drink a fairly standard amount of water each day, but there is still variability in the amount of intake. It would be more accurate if penicillin amount was provided as the concentration in the water and not the dose per mouse.

2. Were lactobacilli detected in the stool of treated mice?

3. Statistical analyses were not provided for sequencing data analysis. Was community structure significantly different, and were relative abundances of specific taxa significantly different?

4. In general, there is little information provided for the sequencing data (e.g., sequencing depths, quality and filtering strategies, etc.). It is also not clear why sequencing data was not split based on gender since there were gender differences in behavior.

5. The wording pertaining to changes in relative abundances of certain bacterial types needs to be revised. For example, changes in relative abundances (based on proportions of bacteria) are not the same as changes in abundance (based on absolute number of bacteria).

6. It is not clear what sub-threshold social defeat is. Can the authors provide more information on this?

7. In several places it is stated that neurochemistry was measured. It is not clear what this means. Do the authors mean neuroinflammation or do they mean expression of the arginine vasopressin receptor?

Reviewer #3 (Remarks to the Author):

In this manuscript, the authors utilized a murine model to investigate the effects of prolonged, low-dose, early-life penicillin exposure on immune (i.e., inflammatory markers), brain (i.e., neurochemistry and blood brain barrier), and behavioral traits. They found that antibiotic-treated mice exhibited altered microbiome composition and brain neurochemistry, as well as impaired anxiety-like and social behaviors and increased aggression, relative to control animals. The authors also found that concurrent supplementation with *Lactobacillus rhamnosis* JB-1, a known psychoactive probiotic, blunted and/or prevented these effects.

Given the frequent use of antibiotics in children and concerns regarding the potential long(er)-term effects on health, these data are intriguing with respect to their observations of altered brain chemistry and behavior. They are also encouraging, with the implication that probiotic

supplementation may serve to blunt or prevent these effects.

Although the authors present a very compelling body of work, I feel that the manuscript could be improved by consideration of the following:

1) In the abstract, the authors suggest that supplementation with *Lactobacillus rhamnosis* JB-1 restored certain biological and behavioral alterations. Restoration suggests that these changes occurred and were subsequently corrected. Given the design of the study (i.e., concurrent exposure with antibiotics and the probiotic) and its outcomes, the data appear to suggest that these effects were prevented and/or blunted, rather than restored.

2) Although the authors sought to improve over previous antibiotic exposure studies by providing a clinically-relevant dose of penicillin and should be congratulated for this, one wonders to what degree the duration of exposure is also clinically relevant and would encourage the authors to address this point.

3) Key information regarding stool sample collection and bioinformatics steps (e.g., which quality filtering thresholds, read-joining, and OTU clustering algorithms were used?) is missing.

4) I would encourage the authors to deposit their sequence data into a public archive. There is no mention of sequence accession numbers in the manuscript.

5) The authors used parametric statistics (repeated measures ANOVA) in their analysis of microbiome data over time. Did the data meet the assumptions of normality and equality of variance?

6) Pg 23, RNA and RT-qPCR methods – Please describe the genes (or at least types of genes) that were evaluated for differential expression.

7) Pg 10, line 188 – Why was the unweighted UniFrac metric chosen to display this data, particularly in light of the discussion of the relative abundances of Proteobacteria that follow the reporting of the UniFrac-based results?

8) Figure 2 (and most others) --- please indicate the number of animals represented in each group and/or comparison.

9) Figure 3 – The same amount of variation is explained along the axes of the three PCoA plots. Given that the three plots are presented to illustrate independent points, they should also be constrained specifically to the data contained within each (i.e., each PCoA should be specific to the data it contains, not sub-plotted from the analysis of data points contained in each of the three frames). This applies to Figure 4A as well.

10) Pg 12, line 241 – Increases in *Avpr1b* expression are described as being increased in the frontal cortex but not hippocampus. The corresponding figure (Fig 6A/B) does not include the hippocampal data.

Point-by-point response to the referee's comments

Reviewer #1

- 1) *It would seem that the study design actually includes a number of insults to the developmental trajectory of the early life microbiota. As the authors note, the maternal microbiome is also altered and there is likely differential postnatal microbiota transmission as well as antibiotic effects. This deviates quite a bit from the study by Cox et al, cited as inspiration for this protocol, in which the low dose penicillin was delivered from birth. I am curious also as to whether the different treatments impacted on maternal care? Was the dams physiology altered as well as their microbiota and could this have impacted in utero? The design also does not permit analysis of the impact of the lactobacillus strain on it's own, something I don't believe has been assessed previously in a similar context.*

Response:

We actually have followed the design of Cox et al.¹ showing that early life low-dose penicillin induced long-term metabolic alterations and change in body composition. While the authors stated in their abstract that penicillin treatment was begun at birth one should pay particular attention to the description of the methods in the supplemental data. Indeed, in the experiments described in this article, the penicillin treatment was given to the pregnant dams 1 week before delivery, and not at birth as written in the abstract. To avoid further confusion, we personally contacted the senior authors of the paper (Blaser and Cox) to clarify their experimental design and they have kindly confirmed that antibiotic treatment was started an estimated 5-7 days before pups' delivery and continued throughout the nursing period and conceded that the abstract has been misstated. They stated that they selected this treatment window to induce maternal microbiota alterations before birth so that the offspring would be colonized with an altered microbiota. Nevertheless, we agree it would be very useful to repeat the experiment by treating separate cohorts of mice with penicillin from birth to weaning or only in utero to establish whether pre- or postnatal-only treatment shed clues on the importance of timing of treatment in the development of a specific adult phenotype. We are now testing this question in current experiments. These perspectives have been added in the discussion section (page 15).

We agree with the reviewer that maternal care could play a role in shaping the behavior in adulthood. Previous reports by Sudo and colleagues², who assessed maternal care in germ-

free (GF) animals, did not find any effect of GF status. Maternal care was not assessed in this current study. Since the treatment was given during a very sensitive developmental period for the pups, we did not want to add maternal stress by disturbing them with an assessment of maternal care. However, nests were carefully checked (twice a week: during the period of food and water change and while weighing the dams) and no abnormalities or cannibalism occurred, suggesting that intestinal dysbiosis did not affect maternal care. This has now been added to the text (discussion section, page 17).

The second research question in this study was to check whether a supplementation with *Lactobacillus rhamnosus* JB-1, which has been shown to have neuroactive properties in healthy adult mice³, could counteract the behavioral and biological alterations induced by early life antibiotic treatment. We wished to test in a separate group whether any potentially deleterious effects of antibiotic therapy might be partially ameliorated by administration of a beneficial bacteria. Indeed this was the case. We believe that our current data now support the possibility that ingestion of probiotic bacteria or possibly prebiotics concurrently with an antibiotic, if they have to be given early in life, might well be beneficial. We are now performing experiments seeking to find out what the effects of JB-1 alone are on the various parameters explored in our study.

2) The authors consider the dose of antibiotic more clinically relevant. This might be the case but this equivalence is somewhat lost in the chronic sustained treatment. The introduction and discussion should really note this important caveat.

Response:

We closely followed the model described in Cox et al¹ that clearly showed effects on metabolism and body weight. In that model, low-dose penicillin was given continuously from before birth to weaning or later. Other models of continuous exposure to antibiotics during early life have shown immunological effects and increased susceptibility to allergic and inflammatory diseases⁴. Previous papers have investigated the effect of antibiotics on behavior but those studies were performed by using a cocktail of large doses of antibiotics (eg ampicillin, vancomycin, neomycin, metronidazole, amphotericin-B) in adults and not in the early life period. Since all the metabolic, immunological and microbial disturbances using continuous exposure to antibiotics early in life have been deeply analyzed by others, we decided for our current study to use a low (clinical) sustained dose of a single very commonly

used pediatric antibiotic (penicillin) which is more clinically relevant to the question we have posed. Nevertheless, we agree with the reviewer that sustained exposure may not mimic completely the common use of antibiotics in children and other studies should be designed to test the most relevant and clinically translatable number and duration of repeated exposures of antibiotics, including recovery phases, during early life. We have added this comment in the discussion (page 15).

3) The question being asked here is clearly important and justified but the introduction meanders a bit and would benefit from a more streamlined revision

Response:

We thank the reviewer for highlighting the importance of the research question. We have modified some paragraphs of the introduction to make the scope of the paper more understandable.

4) Despite some of the reservations noted in (1) above, the intervention produces robust alterations at the behavioral and microbiota level. However, I find the biological analyses a little difficult to interpret since as far as I understand, there is no assessment in stress-naïve male animals. Although the authors do point out that the male and female animals are analysed separately because of this, the results in the male animals may well be a result of a stress of social defeat X antibiotic interaction. Either alone are known to alter the microbiota as well as many of the biological features reported (e.g. BBB, intestinal permeability). Was the microbiota assessed at the end of the behavioural testing?

Response:

First, we have, in the revised manuscript, for clarity replaced the term “sub-threshold defeat” with the alternate term used in the literature, “microdefeat”^(5,6). The microdefeat has been indeed performed in all males (but not in females) and there is no biological assessment in stress-naïve male animals. We could not use females in this test because the aggressor is a male. There are no currently widely accepted versions of social defeat in females. All behavioral tests and intestinal microbiota analysis in males have been performed before social defeat and consequently the results are not influenced by a potential stress effect. The figure

of the study design (Figure 1) has been slightly changed to insert the microdefeat procedure. The microdefeat (3 resident-intruder sessions of 3 min within the same day) has been used before in C57BL/6 mice to measure increased susceptibility to stress and under control conditions, this protocol did not result in social avoidance. To our knowledge, microdefeat has not been previously described in other than C57BL/6 mice strains such as BALB/c and we have therefore no overview of the potential biological alterations induced by this protocol. Regardless, using this microdefeat protocol, we found that under control conditions (naïve BALB/c), almost all mice display social interaction impairment (social avoidance) after 24h. Almost half of antibiotic-treated mice interacted with the aggressor CD-1 mouse which they unusually attacked, at this time point. Regarding the biological parameters that could be influenced by the stress procedure, we measured the serum corticosterone levels (a marker of stress and activation of HPA axis) and intestinal permeability.

We found a significant main effect of sex ($F_{1,57} = 30.58, P < 0.001$) but no effect of treatment ($F_{2,57} = 0.19, P = 0.83$) and no interaction treatment*sex ($F_{2,57} = 1.32, P = 0.28$) meaning that female mice had higher levels of CORT than male mice, although males were subjected to social defeat stress while females were not stressed. However, AB treatment did not influence the level of CORT, suggesting that early life AB exposure had no effect on long-term HPA axis activation. This is in line with another study reporting no effect of AB treatment during adolescence on CORT levels (Desbonnet et al, 2015). The difference in CORT levels between males and females has been previously described in the literature and is a common sex-associated feature for most mammalian species^{7,8}. Even if we do not have here CORT levels in stress-naïve males, data in a previous paper³ using the same method and same kit (ELISA with detection range of 32 to 20,000 pg/mL) for measuring serum CORT in adult BALB/c males, suggest that defeated males in our study have levels of CORT in the same range as non-stressed males (< 50 ng/ml) but lower levels than stressed males (> 200 ng/mL).

In addition, although stress is known to increase intestinal permeability, we did not find significant differences in the fecal albumin level between (stressed) males and (non-stressed) females (sex effect : $F_{1,54} = 2.0$, $P = 0.16$). However, a significant treatment effect ($F_{2,54} = 7.48$, $P = 0.001$) was observed due to the fact that AB-treated male mice had lower fecal albumin level than control male mice, as reported in our manuscript.

These results may suggest that microdefeat in BALB/c mice does not induce short-term (24h later) biological alterations, and consequently, the alteration of the BBB and brain cytokine changes may result from early life antibiotic treatment. This hypothesis is also supported by the fact that non-stressed antibiotic treated females exhibited a very similar phenotype to stressed males with regard to BBB and brain cytokines changes.

- 5) *The manuscript states that mice were sacrificed by decapitation the day following the last test in order to collect trunk blood and tissue samples. Does this mean that the females (who were not assessed in the social defeat paradigm) were sacrificed at a different time point to the male animals? Or that they had a greater recovery time following the last test if sacrificed at a time point equivalent to the male animals?*

Response:

Male and female mice were subjected to the same battery of behavioral tests (with 2 days of rest between each test) which ends with the elevated plus maze (EPM). Females were sacrificed the day following EPM. After EPM, males were subjected to microdefeat (day 1) and tested for social avoidance (day 2), then sacrificed the day following the social avoidance test (day 3). Males were therefore sacrificed a few days after the females, but both were sacrificed the day following a behavioral test. Therefore, there was no extra recovery time for

the females. The figure of the study design (Figure 1) has been slightly changed to insert the microdefeat paradigm in males.

6) *The authors report that although there was no treatment effect on tight junction protein expression in the frontal cortex, they were positively correlated with neuroinflammation in both males and females. I am not convinced of the relevance of this finding – perhaps a revised manuscript could elaborate on this point since I'm not sure how an intact BBB could be linked to the observed inflammatory alterations?*

We agree with the reviewer that the links between BBB integrity and cytokines changes are intriguing. Many cytokines have been investigated in regard to epithelial/endothelial function⁹. While TNF α and IL-1 β are associated with a breakdown of the BBB in multiple models, injection of anti-inflammatory TGF- β has been reported to promote endothelial permeability of the SNC. In the hippocampus, we found an up-regulation of occludin and claudin-5 expression of the BBB and no significant change in any cytokines. Hippocampal tight junction (TJ) expression was negatively correlated with cytokine expression meaning that the increased TJ expression is associated with the absence of cytokines. However, in the frontal cortex, increased cytokines IL-6, IL-10 and Cxcl15 (IL-8) were positively correlated with TJ, which could suggest an anti-inflammatory role of these cytokines. We have now avoided the term neuroinflammation as the findings are only in keeping with a cytokine response and we did not see the classic pro-inflammatory response of IL-1 β or TNF α (see below).

Correlations between tight junction expression and inflammatory cytokine IL-6 in the hippocampus and frontal cortex of male mice. R = Pearson's r coefficient.

Several papers by the Sheridan group have shown an increase in myeloid cells trafficking to the brain (Ly6C^{hi} monocytes) as well as in brain macrophages associated with chronic social stress responses and anxiety-like behavior¹⁰ which was also associated with increased brain expression of IL-1 β , TNF α and reduced IL-10^{11,12}. Moreover, the importance of IL-1 signaling was revealed by studies showing that social stress did not increase the level of circulating monocytes trafficking to the brain in mice knock-down for IL-1 receptor 1, and anxiety-like behavior did not develop in those mice¹³.

However, Möhle et al¹⁴ recently found that antibiotic treatment decreased the number of brain Ly6C^{hi} monocytes, which was associated with decreased neurogenesis in the hippocampus. Supplementation with a mixture of probiotics (VSL#3) and physical exercise were able to restore hippocampal neurogenesis and increase the number of Ly6C^{hi} monocytes in the brain, suggesting that this cell population serves as an important mediator between the periphery and the brain and is crucial for brain homeostasis. Consequently, increased influx of monocytes in the brain was considered by these authors to be a beneficial response to ensure brain homeostasis. Other studies^{15,16} have emphasized, in multiple mouse models of brain diseases, the neuroprotective functions of bone marrow-derived microglia which, by contrast to resident microglia that produced pronounced levels of pro-inflammatory cytokines (TNF α , IL-1 β), can become resolving cells and secrete anti-inflammatory cytokines.

We did not assess the number of brain monocytes in our study. The cytokines that were found to be increased in our study in the frontal cortex were IL-6, IL-10 and Cxcl15 (or IL-8) while TNF α and IL-1 β , the two main markers of the pro-inflammatory status, were not up-regulated. IL-6 is known to have a dual effect in the brain with pro- and anti-inflammatory properties, IL-10 is a typical anti-inflammatory cytokines and has been shown to down-regulate TNF α and IL-1 β in the brain, and IL-8 stimulates the production of neurotrophic factors.

We have now altered the text in the discussion (pages 19-21) and tried to steer clear of describing cytokine changes as "inflammatory". We speculate that the change in cytokines that we found in the frontal cortex of antibiotic treated mice does not mimic a typical pro-inflammatory status but could be a specific response of the brain to cope with early life antibiotic treatment. We are now studying this question of brain cytokines changes in more detail.

7) It is stated that qPCR was used to show that *L. rhamnosus* was detectable in feces of dams that received both AB and JB-1. I would argue that it would be more important to determine whether AB treatment affected the abundance of JB-1 compared to JB-1 treatment alone. I note as well that the protocol cited by the authors (Wang et al.) in support of limiting the adverse effect of penicillin on JB-1 was conducted and validated using a different strain and may not generalise.

We indeed performed qPCR analysis to detect the quantity of *Lactobacillus rhamnosus* in the dams of the 3 treatments (Tx) groups and at 3 different times (T0: before Tx, T1: after 1 week of Tx and T2: after 4 weeks of Tx). We found that *L. rhamnosus* was low but detectable in all groups before the treatment, then detectable, at T1 and T2, only in dams that receive the supplementation with *L. rhamnosus* JB-1. This supports the facts that even if penicillin kills Gram positive bacteria (including *Lactobacillus*), the protocol using two separated bottles of water alternately (1 bottle for penV during the night and 1 bottle for JB-1 during the day) proves to be efficient in maintaining the presence of some *L. rhamnosus* in the gut. This information has been added in the method section (page 23). By using a similar protocol, Wang et al¹⁷ found that ampicillin (which is also a penicillin) treatment alone led to a significant reduction of *Lactobacillus* while treatment of rats with ampicillin + *Lactobacillus fermentum* strain NS9 restored a higher level of *Lactobacillus*.

T0 = before Tx
T1 = after 1 week of Tx
T2 = after 4 weeks of Tx

In the offspring, the genus *Lactobacillus* was present and detectable in all control mice at 3- and 6-weeks old. At PND21, the control microbiota was characterized by a higher abundance of *Lactobacillus*, which decreased as mice mature. However, the levels were very low (<0.1%) in AB-treated mice at both times and supplementation with JB-1 did not significantly

increase the relative abundance of Lactobacillus compared to AB mice. This table has been added in the supplemental (Table S1).

	CT		AB		AB/JB1	
Relative abundance (%)	PND21	PND42	PND21	PND42	PND21	PND42
Lactobacillaceae (family)	7.1	1.7	0.1	0.01	0.01	0.1
Lactobacillus (genus)	6.7	1.6	0.1	0.0	0.0	0.1

8) *Any revised submission should include measures of alpha diversity of the microbiota.*

We thank the reviewer for this important comment. We have now included in the revised manuscript the alpha diversity measurements (Supplemental Figures 7 and 9) and added the following to the text (results section - page 9):

DAMS: When looking at the alpha diversity (Fig S7) at T0 there was no difference between the groups but at T1 and T2 the AB and AB/JB1 groups had lower diversity compared to controls. Due to the small number of animals (n = 3-4 dams/groups), the results did not reach the statistical significance but the lack of overlap between boxplot clearly indicates that antibiotic treatment reduced alpha diversity.

Fig: Boxplots for community richness (α -diversity) in dams at three different time point. **A.** before AB treatment (T0); **B.** one week after AB treatment (day of delivery, T1); **C.** four weeks after AB treatment (day of weaning). Data shown are Faith's phylogenetic diversity (PD)

OFFSPRING: The diversity measures of the offspring showed a strong AB effect which caused the AB and AB/JB1 groups to be significantly less diverse than the control group (Fig. S9) at both time points (PND21 and PND42).

Fig. Boxplots for community richness (α -diversity) in pups at two time points: postnatal day 21 (weaning) (A) and postnatal day 42 (B). Data shown are Faith's phylogenetic diversity (PD). ** $P < 0.01$ vs CT.

- 9) *For the offspring, there were only 6 male animals on the combined JB-1/AB intervention. The possibility that this resulted in an underpowered analysis should be noted as a limitation in the discussion.*

Response:

Unfortunately, the last litter treated with AB/JB-1 contained 8 pups: 6 females and only 2 males. One male was born without tail so we decided not to use it for our experiments. The last male remaining was not used for the experiments as it would have been housed singly which is considered to induce social isolation and stress that was not compatible with behavioral assessment. That actually explains the lower number of males in this group.

While the restoration of certain biological and behavioral results was statistically significant with only 6 males, we agree with the reviewer that the partial restoration of anxiety-like behavior in the AB/JB-1 male group could have been significant with more animals, or by contrast, reduced. Similarly, the expression of tight junctions in the BBB of AB/JB-1 males was in between the level of controls and the level of AB males. The phenotype of AB/JB-1 would have been clearer and we would have gained statistical power with more animals. We have added this limitation in the discussion section (page 15).

10) Could the authors comment in the revised submission on the value of a microbiota transfer to resolve the question of whether it is the altered microbiota or the antibiotics that induced the observed alterations?

We do not think that the behavioral alterations could be attributable to a direct toxic effect of penicillin for several reasons: 1) penetration of penicillin into the CSF is very low in the absence of infection¹⁸ (see the recent paper of Fröhlich et al¹⁹ showing that ampicillin does not reach the brain following gavage) and 2) the renal clearance of penV is very rapid (approximately 20–40% of an oral dose is excreted in urine within 6 hours) and AB treatment was stopped 3w before beginning behavioral assessment.

Consequently, we think that alteration of the gut microbiota induced by AB treatment may be the driver of behavioral changes. To eliminate the direct effect of penicillin on the host, we agree with reviewer that we would need to transfer the fecal microbiota of AB-treated or control mice to germ-free mice and subsequently submit them to behavioral testing. This approach has been used to test and possibly support a causal role of altered microbiota. Cox et al¹ transferred the gut microbiota of mice treated with low dose penicillin to germ-free mice and showed that the induced metabolic and immunological alterations were transferrable to germ-free recipients, demonstrating that altered microbiota, and not antibiotics per se, likely played a causal role. However, a recent careful critique of the procedure of transplantation of fecal material to germfree hosts²⁰ outlined some important problems. Engraftment of fecal microbiota to germ-free mice results in only a partial resemblance to the donor microbiota as predominant taxa of the gut microbiome fail to efficiently colonize the gut. This is particularly true for human to mouse engraftment but also applies to mouse to mouse experiments. We will however attempt these experiments in the future. This comment on microbiota transfer has been added in the third section of the discussion (page 17).

11) Am I correct in thinking that the microbiota analyses reported for the offspring in the supplementary data does not include a breakdown according to sex?

Response:

We did not find significant difference in gut microbiota composition between males and females that is why we presented the figures of β -diversity with both sexes. However, for the

interest of the reviewer, here are the PCoA analyses of β -diversity distinguishing males and females by different colors. These graphs have been inserted in the supplemental (Fig S15) (page 10).

Regarding the relative abundance, the only difference found concerns the family *Anaeroplasmataceae* (phylum Tenericutes/ class Mollicutes). Analysis revealed significant time* treatment ($F_{(2,60)} = 5.2, P = 0.009$) and time*sex interaction ($F_{(1,60)} = 10.52, P = 0.002$) suggesting that the change of the level of this bacteria from weaning to 6-week old is different between treatment groups as well as between male and female mice. To our knowledge, the impact of *Anaeroplasmataceae* on host physiology remains unknown. This results has been added in the supplemental data (Fig S16) (page 10).

12) The discussion would benefit from a more detailed consideration of the sex-specific behavioral alterations in contrast to the similarities for the neuroinflammation data.

Response:

While the gut microbial structure was similar between male and female mice, we found some differences regarding the behavioral data. For instance, early life antibiotic treatment induced a decrease in anxiety-like behavior in males but not in females, while the concurrent supplementation with *L.rhamnosus* JB-1 decreased anxiety in females and restored normal anxiety level in males. In addition, the reduction of social behavior induced by antibiotic treatment was more pronounced in females than males. It has been previously reported that treatment with *L.rhamnosus* JB-1 induces an anxiolytic effect through the stimulation of the vagus nerve in adult male mice. Unfortunately, no data are available on this effect of *L.rhamnosus* in females.

A previous study²¹, performed in germ-free mice, showed that CNS alterations (serotonergic alterations, BDNF levels) occurred in a sex-specific manner, while peripheral immunological and neuroendocrine (corticosterone release following acute stressor) measures were similar in both sexes. Reconstitution of a normal microbiota restored anxiety-like behavior in males but not in females. In addition, anxiety-like behavior can be mouse-strain dependent²².

The mechanisms surrounding these sex differences are not well understood, but could be influenced by estrous cycle hormones. However, the short duration of estrus in mice, the longer duration of the behavioral testing period as well as the synchronization of estrous cycle that could occur in group housed females (Whitten effect), would suggest that the behavioral differences seen between males and females are likely not due to cyclical female sex hormone variation. By contrast, it might be possible that another, yet unidentified, immunological or neuroendocrine factor or bacterial metabolites could be at the origin of sex differences in the behavioral response to antibiotic and JB-1 treatments. Consequently, fecal and blood metabolomics analysis should be addressed in future studies to explore this possibility. Moreover, the influence of the vagus nerve in the gut-brain communication and its potential sex-difference effect should also be addressed in this context of early life antibiotic treatment in the future.

Altogether, these results reinforce the importance of investigating sex- (and strain) differences in rodent studies since results obtained in males may not be generalized in females, as it was recently reported in a study showing impressive sex-differences in neuro-immune signaling²³. Also, the gender difference should be considered an opportunity to investigate more deeply brain function and behavior rather than an unwanted complication of future research.

A shorter version of this comment has been added in the discussion section (pages 17-18).

References:

1. Cox, L. M. *et al.* Altering the intestinal microbiota during a critical developmental window has lasting metabolic consequences. *Cell* **158**, 705–721 (2014).
2. Sudo, N. *et al.* Postnatal microbial colonization programs the hypothalamic-pituitary-adrenal system for stress response in mice. *J. Physiol.* **558**, 263–275 (2004).
3. Bravo, J. A. *et al.* Ingestion of Lactobacillus strain regulates emotional behavior and central GABA receptor expression in a mouse via the vagus nerve. *Proc. Natl. Acad. Sci. U. S. A.* **108**, 16050–16055 (2011).
4. Russell, S. L. *et al.* Early life antibiotic-driven changes in microbiota enhance susceptibility to allergic asthma. *EMBO Rep.* **13**, 440–447 (2012).
5. Golden, S. A. *et al.* Epigenetic regulation of RAC1 induces synaptic remodeling in stress disorders and depression. *Nat. Med.* **19**, 337–344 (2013).
6. Krishnan, V. *et al.* Molecular Adaptations Underlying Susceptibility and Resistance to Social Defeat in Brain Reward Regions. *Cell* **131**, 391–404 (2007).
7. Reeder, D. M. & Kramer, K. M. stress in free-ranging mammals: integrating physiology, ecology and natural history *J. Mammal.* **86**, 225–235 (2005).
8. Malisch, J. L. *et al.* Baseline and Stress-Induced Plasma Corticosterone Concentrations of Mice Selectively Bred for High Voluntary Wheel Running. *Physiol. Biochem. Zool.* **80**, 146–156 (2007).
9. Gloor, S. M. *et al.* Molecular and cellular permeability control at the blood-brain barrier. *Brain Res. Brain Res. Rev.* **36**, 258–264 (2001).
10. Wohleb, E. S., Powell, N. D., Godbout, J. P. & Sheridan, J. F. Stress-induced recruitment of bone marrow-derived monocytes to the brain promotes anxiety-like behavior. *J. Neurosci. Off. J. Soc. Neurosci.* **33**, 13820–13833 (2013).
11. Voorhees, J. L. *et al.* Prolonged restraint stress increases IL-6, reduces IL-10, and causes persistent depressive-like behavior that is reversed by recombinant IL-10. *PloS One* **8**, e58488 (2013).

12. Sawicki, C. M. *et al.* Social defeat promotes a reactive endothelium in a brain region-dependent manner with increased expression of key adhesion molecules, selectins and chemokines associated with the recruitment of myeloid cells to the brain. *Neuroscience* **302**, 151–164 (2015).
13. Wohleb, E. S. *et al.* Knockdown of interleukin-1 receptor type-1 on endothelial cells attenuated stress-induced neuroinflammation and prevented anxiety-like behavior. *J. Neurosci. Off. J. Soc. Neurosci.* **34**, 2583–2591 (2014).
14. Möhle, L. *et al.* Ly6Chi Monocytes Provide a Link between Antibiotic-Induced Changes in Gut Microbiota and Adult Hippocampal Neurogenesis. *Cell Rep.* **15**, 1945–1956 (2016).
15. Naert, G. & Rivest, S. A deficiency in CCR2⁺ monocytes: the hidden side of Alzheimer’s disease. *J. Mol. Cell Biol.* **5**, 284–293 (2013).
16. Shechter, R. & Schwartz, M. Harnessing monocyte-derived macrophages to control central nervous system pathologies: no longer ‘if’ but ‘how’. *J. Pathol.* **229**, 332–346 (2013).
17. Wang, T. *et al.* Lactobacillus fermentum NS9 restores the antibiotic induced physiological and psychological abnormalities in rats. *Benef. Microbes* **6**, 707–717 (2015).
18. Grayson, M. L. *et al.* *Kucers’ The Use of Antibiotics Sixth Edition: A Clinical Review of Antibacterial, Antifungal and Antiviral Drugs.* (CRC Press, 2010).
19. Fröhlich, E. E. *et al.* Cognitive impairment by antibiotic-induced gut dysbiosis: Analysis of gut microbiota-brain communication. *Brain. Behav. Immun.* **56**, 140–155 (2016).
20. Arrieta, M.-C., Walter, J. & Finlay, B. B. Human Microbiota-Associated Mice: A Model with Challenges. *Cell Host Microbe* **19**, 575–578 (2016).
21. Clarke, G. *et al.* The microbiome-gut-brain axis during early life regulates the hippocampal serotonergic system in a sex-dependent manner. *Mol. Psychiatry* **18**, 666–673 (2013).
22. Holmes, A., Li, Q., Murphy, D. L., Gold, E. & Crawley, J. N. Abnormal anxiety-related behavior in serotonin transporter null mutant mice: the influence of genetic background. *Genes Brain Behav.* **2**, 365–380 (2003).

23. Sorge, R. E. *et al.* Different immune cells mediate mechanical pain hypersensitivity in male and female mice. *Nat. Neurosci.* **18**, 1081–1083 (2015).

Reviewer #2

Major comments

- 1) *There seems to be a control group missing (namely no antibiotic, but with JB1). Please justify why a fully crossed 2 x 2 experimental design (control vs. antibiotic x vehicle vs. JB1) was not used.*

Response:

The first aim of this study was to investigate whether early life antibiotic exposure could induce long-term biological and behavioral alterations. The second research question was to check whether a supplementation with *Lactobacillus rhamnosus* JB-1, which has been shown to have neuroactive properties in healthy adult mice¹, could counteract the behavioral and biological alterations induced by early life antibiotic treatment. We wished to test in a separate group whether any potentially deleterious effects of antibiotic therapy might be partially ameliorated by administration of a beneficial bacteria. Indeed this was the case. We believe that our current data now support the possibility that ingestion of probiotic bacteria or possibly prebiotics concurrently with an antibiotic, if they have to be given early in life, might well be beneficial. We are now performing experiments seeking to find out what the effects of JB-1 alone, are on the various parameters explored in our study.

- 2) *The sample size is quite low. Although there is a total of 72 pups (which is a good sample size), only 5 dams were in the control condition, 4 dams were antibiotic treated, and 3 dams were given probiotic.*

Response:

We agree with the reviewer that the dam number per treatment group is relatively small. In ideal conditions, the sample size should be based on the number of dams (or litters) and not on the number of pups. Moreover, litters should be broken up and randomly assigned to the different treatment groups. However, as the dams were the animals treated and not the pups, it was impossible to randomly assign the pups to a treatment group. We acknowledge that the relatively small number of litters is a potential shortcoming of the study and we have added this limitation in the discussion section (page 15) as well as in the method section (page 22).

3) *It is not entirely clear what the real purpose/conclusion of the study is. As it is currently written, the manuscript is really just a description of the effects of prenatal/early life antibiotics on mouse biology without placing the results into context. For example, the first paragraph of the discussion indicates that penicillin AB are frequently prescribed in early life, but their effects on adulthood diseases such as allergies, IBD, and obesity are not known. Why does this study focus on the brain and behavior and not the aforementioned diseases? Is there evidence that early life antibiotics lead to mental illness later in life?*

Response:

As reported by Chai et al², antibiotics are the most frequently prescribed drugs in pediatric patients. Epidemiological studies (in humans) have revealed an association between early life antibiotic treatment and an increased risk of developing allergies, inflammatory bowel diseases and obesity. Experimental models of allergies and obesity (in rodents) have demonstrated that intestinal dysbiosis induced by early life antibiotic exposure play a causal role in the development and long-term persistence of metabolic and immunological alterations. Given 1) the frequent exposure of children to antibiotics, 2) the high rate of mental disorders in adults (47% (lifetime prevalence) of the population in the USA suffer from anxiety, mood or substance disorders)³ and 3) the major influence of gut microbiota on brain and behavior, the goal of our study was to investigate whether early life dysbiosis could also induce long-term behavioral alterations. This seems to be particularly important at the present time when the literature is replete with evidence that the gut microbiome has profound effects on the development and maturation of the nervous system and many reviews have appeared supporting the unproven suggestion that dysbiosis may be the cause of many psychiatric illnesses.

We have modified some paragraphs of the introduction to make the scope of the paper more understandable.

Recently, a large population-based study⁴ showed for the first time an association between exposure to antibiotics and higher risk for depression, anxiety but not psychosis in adults. The risk increased with the number of antibiotic exposures (higher risk for recurrent exposures). The odds ratios revealed significant but weak associations. In addition, another very recent study⁵ has demonstrated an association between antibiotic treatment in the first year of life and poorer cognitive, behavioral and emotional outcomes throughout childhood (at 3.5, 7 and

11 years old). Unfortunately, information on dosage, duration, types of antibiotic and reason for antibiotic use was not collected. In these studies, we do not know whether antibiotics, infections or other factors associated with the reasons for prescription of antibiotics are the potential causes for elevated risk of emotional and cognitive disturbances.

4) Related to the point above, how do the authors interpret the increased neuroinflammation that has been shown by multiple groups to lead to increased anxiety in mouse models? Why do the animals in this study have reduced anxiety, but increased neuroinflammation?

Response:

We agree with the reviewer that in multiple models, increased brain inflammation has been associated with increased anxiety-like behavior. Increased anxiety associated with neuroinflammation was mainly induced by social stress⁶, LPS injection⁷, IL-1 β intracerebroventricular injection⁸, parasite infection⁹, alcohol¹⁰, etc. In all these models, pro-inflammatory cytokines TNF α and IL-1 β were up-regulated and could participate to increased anxiety. In our current study, we did not see an increase in TNF α and IL-1 β . We have now avoided in the revised manuscript the use of the term “inflammation” as the cytokine responses we have observed may have protective implications (see below).

Several papers by the Sheridan group have shown an increase in myeloid cells trafficking to the brain (Ly6C^{hi} monocytes) as well as in brain macrophages associated with chronic social stress responses and anxiety-like behavior¹¹, which were also associated with increased brain expression of IL-1 β , TNF α and reduced IL-10^{6,12}. The importance of IL-1 signaling was revealed by studies showing that social stress did not increase the level of circulating monocytes trafficking to the brain in mice knock-down for IL-1 receptor 1, and anxiety-like behavior did not develop in those mice¹³.

However, Möhle et al¹⁴ recently found that prolonged antibiotic treatment decreased the number of brain Ly6C^{hi} monocytes, which was associated with decreased neurogenesis in the hippocampus. Supplementation with a mixture of probiotics (VSL#3) as well as physical exercise were able to restore hippocampal neurogenesis and increase the number of Ly6C^{hi} monocytes in the brain, suggesting that this cell population serves as an important mediator between the periphery and the brain and is crucial for brain homeostasis. Consequently,

increased influx of monocytes in the brain was considered by these authors to be a beneficial response to ensure brain homeostasis. Other studies^{15,16} have emphasized, in multiple mouse models of brain diseases, the neuroprotective functions of bone marrow-derived microglia which, by contrast to resident microglia that produced pronounced levels of pro-inflammatory cytokines (TNF α , IL-1 β), can become resolving cells and secrete anti-inflammatory cytokines. We did not assess the number of Ly6C^h monocytes in our study. The cytokines that were found to be increased in our study in the frontal cortex were IL-6, IL-10 and Cxcl15 (or IL-8) while TNF α and IL-1 β , the two main markers of the pro-inflammatory response, were not up-regulated. IL-6 is known to have a dual effect in the brain with pro- and anti-inflammatory properties, IL-10 is a typical anti-inflammatory cytokines and has been shown to down-regulate TNF α and IL-1 β in the brain, and IL-8 stimulates the production of neurotrophic factors. We therefore speculate that the change in cytokines that we found in the frontal cortex of antibiotic treated mice does not mimic a typical pro-inflammatory status but could be a specific response of the brain to cope with early life antibiotic treatment. We have altered the text in the discussion (pages 19-20) to make this possibility clear and tried to steer clear of describing cytokine changes as "inflammatory". We are now studying this question of brain cytokines changes in more detail.

In addition, in our current study, no activation of the peripheral immune response (blood, ileum and colon) was observed in mice treated with AB suggesting that cytokine changes-induced by low dose AB is likely a very localized brain response (page 21). Elevation of brain cytokines levels in AB-treated mice was associated with reduced social behavior, reduced preference for social novelty and surprisingly, reduced anxiety-like behavior. These results may raise the question "is decreased anxiety-like behavior beneficial (or not) to the mice in some way?". In this regard, *Toxoplasma gondii* infection reduced anxiety-like behavior and made rodents more adventurous, less fearful, more attracted by cat odor and therefore more susceptible to be eaten by a predator^{17,18}. However, in humans, *Toxoplasma* infection has been associated with increased prevalence of generalized anxiety disorders¹⁹. These examples highlight the importance of context and ethological considerations in making appropriate analogies between animal and human behaviors.

Specific comments

- 1) *Penicillin was given in the drinking water. Mice drink a fairly standard amount of water each day, but there is still variability in the amount of intake. It would be more accurate if penicillin amount was provided as the concentration in the water and not the dose per mouse.*

Response:

Penicillin was given in the drinking water of the dams. During gestational and lactating periods, water consumption increased dramatically to compensate the increased metabolism. While BALB/c mice, in normal conditions, drink a standard amount of water ($\approx 3-4$ mL/day), the water consumption of pregnant and lactating dams can reach 10-15 mL/day and depends on the stage of pregnancy and on the number of pups in the litter. Therefore, to make sure that all dams received the same amount of penicillin, water consumption and dams weight were measured twice a week and the amount of penicillin diluted in the water was adjusted accordingly. We have added this information in the method section (subsection: *Treatment with antibiotic and L. rhamnosus JB-1TM*) (page 23).

- 2) *Were lactobacilli detected in the stool of treated mice?*

Response:

In the offspring, the genus *Lactobacillus* was present and detectable in all control mice at 3- and 6-weeks old. At PND21, the control microbiota was characterized by a higher abundance of *Lactobacillus*, which decreased as mice mature. However, the levels were very low ($<0.1\%$) in AB-treated mice at both times and supplementation with JB-1 did not significantly increase the level of *Lactobacillus* compared to AB mice. This table has been added in the supplemental information (Table S1).

	CT		AB		AB/JB-1	
Relative abundance (%)	PND21	PND42	PND21	PND42	PND21	PND42
Lactobacillaceae (family)	7.1	1.7	0.1	0.01	0.01	0.1
Lactobacillus (genus)	6.7	1.6	0.1	0.0	0.0	0.1

We also performed a quantification of *Lactobacillus rhamnosus* by qPCR in the dams of the 3 treatments (Tx) groups and at 3 different times (T0: before Tx, T1: after 1 week of Tx and T2: after 4 weeks of Tx). We found that *L. rhamnosus* was low but detectable in all groups before the treatment, then detectable, at T1 and T2, only in dams that receive the supplementation with *L. rhamnosus* JB-1. This supports the facts than even if penicillin kills Gram positive bacteria (including *Lactobacillus*), the protocol using two separated bottles of water alternately (1 bottle for penV during the night and 1 bottle for JB-1 during the day) that we implemented for this current experiment proves to be efficient in maintaining the presence of *L. rhamnosus* in the gut. This information has been added in the method section (page 23).

T0 = before Tx
T1 = after 1 week of Tx
T2 = after 4 weeks of Tx

- 3) *Statistical analyses were not provided for sequencing data analysis. Was community structure significantly different, and were relative abundances of specific taxa significantly different?*

We have now added the *P* values of the community structure analysis (for alpha and beta-diversity indices) to the text which prove to be significantly different (page 9). A detailed statistical analysis of the relative abundance at the phylum and family levels in the offspring was provided in the supplemental data.

- 4) *In general, there is little information provided for the sequencing data (e.g., sequencing depths, quality and filtering strategies, etc.). It is also not clear why sequencing data was not split based on gender since there were gender differences in behavior.*

Response:

We have now changed the method section regarding the 16S rRNA gene sequence analysis to include more details of the technique used, as asked by the reviewer. This paragraph has now been moved to supplemental information.

We did not find significant difference in gut microbiota composition between males and females; that is why we presented the figures of β -diversity with both sexes. However, for the interest of the reviewer, here are the PCoA analyses of β -diversity distinguishing males and females by different colors. These graphs have been added in the supplemental information (Fig S15) (page 10).

Regarding the relative abundance, the only sex difference found concerns the family *Anareoplasmataceae* (phylum Tenericutes/ class Mollicutes). Analysis revealed significant time* treatment interaction ($F_{(2,60)} = 5.2, P = 0.009$ and time*sex interaction ($F_{(1,60)} = 10.52, P = 0.002$) suggesting that the change of the relative abundance of this bacteria from weaning to 6-week old is different between treatment groups as well as between male and female mice. This new figure has been added in the supplemental (Fig S16) (page 10). To our knowledge, the impact of *Anaeroplasmataceae* on host physiology remains unknown.

5) *The wording pertaining to changes in relative abundances of certain bacterial types needs to be revised. For example, changes in relative abundances (based on proportions of bacteria) are not the same as changes in abundance (based on absolute number of bacteria).*

Response:

We thank the reviewer for this appropriate remark and we have now used the term “relative abundance” (as it is based on the proportion of bacteria) in the revised version of the manuscript.

6) *It is not clear what sub-threshold social defeat is. Can the authors provide more information on this?*

Response:

We have, in the revised manuscript, for clarity replaced the term “sub-threshold defeat” with the alternate term used in the literature, microdefeat. The microdefeat model was developed initially to measure increased susceptibility to stress. This paradigm is a short (or acute) version adapted from the chronic social defeat paradigm described by Krishnan et al²⁰ and Golden SA et al²¹ who investigated the molecular mechanism of the psychopathology of stress-related disorders such as depression and PTSD. Microdefeat has been described in C57BL/6 mice and has been recently shown to induce different brain response than the chronic social defeat²². Microdefeat can reveal a susceptible phenotype if an animal’s stress threshold is shifted by the experimental manipulation.

The procedure is the following. First, CD1 aggressors (retired breeders >4-5 months old) were screened for consistent attack latencies (<60 sec on 3 consecutive screening sessions with BALB/c (or same strain as experimental mice) intruders called screeners). CD-1 mice are singly housed throughout the experiment. Successful application of social defeat stress is dependent on appropriate selection of CD-1 mice with consistent levels of aggressive behavior. Within the same day, BALB/c mice are forced to intrude into the space territorialized by the larger and aggressive CD-1 mouse for 3 sessions of 3 min, followed by a rest period 15 min in the home cage between each exposure. This situation leads to intruder aggression and subordination. After the last resident-intruder session, BALB/c mice are housed singly (with free access of food and water) and tested for social avoidance 24h later. During this test, we measured the time spent in the interaction zone when 1) the aggressor CD-1 is absent (empty enclosure) and 2) CD-1 is present. The socially defeated mice are supposed to actively avoid the CD-1 aggressor by spending more time in the corner zones. Under control condition, this microdefeat protocol does not induce social avoidance in C57BL/6. A detailed description of the microdefeat procedure has been now added to the revised version of supplemental material.

In our study, the microdefeat has been performed in all males. We could not use females in this test because the CD-1 aggressor is a male. There are no currently widely accepted versions of social defeat in females. To our knowledge, microdefeat has not been previously described in other than C57BL/6 mice strains such as BALB/c and we have therefore no overview of the potential biological alterations induced by this protocol. Regardless, using this microdefeat protocol, we found that under control conditions (naïve BALB/c), almost all mice display social interaction impairment (social avoidance) after 24h. Almost half of antibiotic-treated mice interacted with the aggressor CD-1 mouse which they unusually attacked, at this time point.

All behavioral tests and intestinal microbiota analysis in males have been performed before microdefeat and consequently the results are not influenced by a potential stress effect. Microdefeat in males has been performed 48h before sacrifice, and there is no biological assessment in stress-naïve males. Regarding the biological parameters that could be influenced by the stress procedure, we measured and reported here the serum corticosterone levels (a marker of stress and activation of HPA axis) and intestinal permeability.

We found a significant main effect of sex ($F_{1,57} = 30.58$, $P < 0.001$) but no effect of treatment ($F_{2,57} = 0.19$, $P = 0.83$) and no interaction treatment*sex ($F_{2,57} = 1.32$, $P = 0.28$) meaning that

female mice had higher levels of CORT than male mice, although males were subjected to social defeat stress while females were not stressed. However, AB treatment did not influence the level of CORT, suggesting that early life AB exposure had no effect on long-term HPA axis activation. This is in line with another study reporting no effect of AB treatment during adolescence on CORT levels²³. The difference in CORT levels between males and females has been previously described in the literature and is a common sex-associated feature for most mammalian species^{24,25}. Even if we do not have here CORT levels in stress-naïve males, data in a previous paper¹ using the same method and same kit (ELISA with detection range of 32 to 20,000 pg/mL) for measuring serum CORT in adult BALB/c males, suggest that defeated males in our study have levels of CORT in the same range as non-stressed males (< 50 ng/ml) but lower levels than stressed males (> 200 ng/mL).

In addition, although stress is known to increase intestinal permeability, we did not find significant differences in the fecal albumin level between (stressed) males and (non-stressed) females (sex effect : $F_{1,54} = 2.0$, $P = 0.16$). However, a significant treatment effect ($F_{2,54} = 7.48$, $P = 0.001$) was observed due to the fact that AB-treated male mice had lower fecal albumin level than control male mice, as reported in our manuscript.

These results may suggest that microdefeat in BALB/c mice does not induce short-term (24h later) biological alterations, and consequently, the alteration of the BBB and brain cytokines changes may result from early life antibiotic treatment. This hypothesis is also supported by the fact that non-stressed antibiotic treated females exhibited a very similar phenotype to stressed males with regard with BBB and brain cytokines changes.

7) *In several places it is stated that neurochemistry was measured. It is not clear what this means. Do the authors mean neuroinflammation or do they mean expression of the arginine vasopressin receptor?*

Response:

We agree with the reviewer that the term neurochemistry is not appropriate due to the fact that neurotransmitters or other neuropeptides that influence the function of neurons have not been analyzed. We have deleted this term in the revised version of the manuscript. In addition, we have replaced the term “neuroinflammation” by “brain cytokines changes” as we think that the cytokines up-regulated (IL-6, IL-10, Cxcl15) in the frontal cortex do not represent a real pro-inflammatory response, but rather a specific response of the brain to cope with early life antibiotic treatment. (see answer comment#4)

References:

1. Bravo, J. A. *et al.* Ingestion of Lactobacillus strain regulates emotional behavior and central GABA receptor expression in a mouse via the vagus nerve. *Proc. Natl. Acad. Sci. U. S. A.* **108**, 16050–16055 (2011).
2. Chai, G. *et al.* Trends of Outpatient Prescription Drug Utilization in US Children, 2002–2010. *Pediatrics* **130**, 23–31 (2012).
3. Kessler, R. C. *et al.* The global burden of mental disorders: an update from the WHO World Mental Health (WMH) surveys. *Epidemiol. Psychiatr. Soc.* **18**, 23–33 (2009).
4. Lurie, I., Yang, Y.-X., Haynes, K., Mamtani, R. & Boursi, B. Antibiotic exposure and the risk for depression, anxiety, or psychosis: a nested case-control study. *J. Clin. Psychiatry* **76**, 1522–1528 (2015).
5. Slykerman, R. F. *et al.* Antibiotics in the first year of life and subsequent neurocognitive outcomes. *Acta Paediatr. Oslo Nor. 1992* (2016). doi:10.1111/apa.13613
6. Voorhees, J. L. *et al.* Prolonged restraint stress increases IL-6, reduces IL-10, and causes persistent depressive-like behavior that is reversed by recombinant IL-10. *PloS One* **8**, e58488 (2013).
7. Mayerhofer, R. *et al.* Diverse action of lipoteichoic acid and lipopolysaccharide on neuroinflammation, blood-brain barrier disruption, and anxiety in mice. *Brain. Behav. Immun.* doi:10.1016/j.bbi.2016.10.011
8. Song, C., Horrobin, D. F. & Leonard, B. E. The Comparison of Changes in Behavior, Neurochemistry, Endocrine, and Immune Functions after Different Routes, Doses and Durations of Administrations of IL-1 β in Rats. *Pharmacopsychiatry* **39**, 88–99 (2006).
9. Guha, S. K. *et al.* Single episode of mild murine malaria induces neuroinflammation, alters microglial profile, impairs adult neurogenesis, and causes deficits in social and anxiety-like behavior. *Brain. Behav. Immun.* **42**, 123–137 (2014).
10. Pascual, M., Baliño, P., Aragón, C. M. G. & Guerri, C. Cytokines and chemokines as biomarkers of ethanol-induced neuroinflammation and anxiety-related behavior: Role of TLR4 and TLR2. *Neuropharmacology* **89**, 352–359 (2015).

11. Wohleb, E. S., Powell, N. D., Godbout, J. P. & Sheridan, J. F. Stress-induced recruitment of bone marrow-derived monocytes to the brain promotes anxiety-like behavior. *J. Neurosci. Off. J. Soc. Neurosci.* **33**, 13820–13833 (2013).
12. Sawicki, C. M. *et al.* Social defeat promotes a reactive endothelium in a brain region-dependent manner with increased expression of key adhesion molecules, selectins and chemokines associated with the recruitment of myeloid cells to the brain. *Neuroscience* **302**, 151–164 (2015).
13. Wohleb, E. S. *et al.* Knockdown of interleukin-1 receptor type-1 on endothelial cells attenuated stress-induced neuroinflammation and prevented anxiety-like behavior. *J. Neurosci. Off. J. Soc. Neurosci.* **34**, 2583–2591 (2014).
14. Möhle, L. *et al.* Ly6Chi Monocytes Provide a Link between Antibiotic-Induced Changes in Gut Microbiota and Adult Hippocampal Neurogenesis. *Cell Rep.* **15**, 1945–1956 (2016).
15. Naert, G. & Rivest, S. A deficiency in CCR2+ monocytes: the hidden side of Alzheimer's disease. *J. Mol. Cell Biol.* **5**, 284–293 (2013).
16. Shechter, R. & Schwartz, M. Harnessing monocyte-derived macrophages to control central nervous system pathologies: no longer 'if' but 'how'. *J. Pathol.* **229**, 332–346 (2013).
17. Berdoy, M., Webster, J. P. & Macdonald, D. W. Fatal attraction in rats infected with *Toxoplasma gondii*. *Proc. R. Soc. B Biol. Sci.* **267**, 1591–1594 (2000).
18. Gonzalez, L. E. *et al.* *Toxoplasma gondii* infection lower anxiety as measured in the plus-maze and social interaction tests in rats: A behavioral analysis. *Behav. Brain Res.* **177**, 70–79 (2007).
19. Markovitz, A. A. *et al.* *Toxoplasma gondii* and anxiety disorders in a community-based sample. *Brain. Behav. Immun.* **43**, 192–197 (2015).
20. Krishnan, V. *et al.* Molecular Adaptations Underlying Susceptibility and Resistance to Social Defeat in Brain Reward Regions. *Cell* **131**, 391–404 (2007).
21. Golden, S. A. *et al.* Epigenetic regulation of RAC1 induces synaptic remodeling in stress disorders and depression. *Nat. Med.* **19**, 337–344 (2013).

22. Donahue, R. J. *et al.* Effects of acute and chronic social defeat stress are differentially mediated by the dynorphin/kappa-opioid receptor system. *Behav. Pharmacol.* **26**, 654–663 (2015).
23. Desbonnet, L. *et al.* Gut microbiota depletion from early adolescence in mice: Implications for brain and behaviour. *Brain. Behav. Immun.* **48**, 165–173 (2015).
24. Reeder, D. M. & Kramer, K. M. stress in free-ranging mammals: integrating physiology, ecology, and natural history. *J. Mammal.* **86**, 225–235 (2005).
25. Malisch, J. L. *et al.* Baseline and Stress-Induced Plasma Corticosterone Concentrations of Mice Selectively Bred for High Voluntary Wheel Running. *Physiol. Biochem. Zool.* **80**, 146–156 (2007).

Reviewer #3

1) In the abstract, the authors suggest that supplementation with Lactobacillus rhamnosus JB-1 restored certain biological and behavioral alterations. Restoration suggests that these changes occurred and were subsequently corrected. Given the design of the study (i.e., concurrent exposure with antibiotics and the probiotic) and its outcomes, the data appear to suggest that these effects were prevented and/or blunted, rather than restored.

We totally agree with the reviewer that the use of the terms “prevented or blunted” is more appropriate than “restored” since different mice have been treated with either penicillin alone, or penicillin + *L. rhamnosus* JB-1. We have adopted these changes in the abstract as well as in the revised version of the whole manuscript.

2) Although the authors sought to improve over previous antibiotic exposure studies by providing a clinically-relevant dose of penicillin and should be congratulated for this, one wonders to what degree the duration of exposure is also clinically relevant and would encourage the authors to address this point.

Response:

We actually followed the model described in Cox et al¹ that clearly showed that low-dose penicillin induces effects on metabolism, immune response and body weight. In that model, low-dose penicillin was given continuously from 1 week before birth to weaning (or later). Despite the statement in their abstract that penicillin treatment was only started postnatally, the method section in their paper clearly state otherwise and this has also been recently independently confirmed personally to us by the senior authors of the paper (Cox and Blaser). Other models of continuous exposure to antibiotics during the perinatal period (both in utero and after birth) have shown long-term immunological effects and increased susceptibility to allergic and inflammatory diseases^{2,3}. Previous papers have investigated the effect of antibiotics on behavior but those studies were performed by using a cocktail of large doses of antibiotics (ampicillin, vancomycin, neomycin, metronidazole, amphotericin-B) which have little relevance to the clinical situation. Since all the metabolic, immunological and microbial disturbances using continuous exposure to antibiotics early in life have been deeply analyzed by others, we decided to apply this design to our current study and use a low (clinical)

sustained dose of a single very commonly used pediatric antibiotic (penicillin) which is more clinically relevant to the question we have posed. Nevertheless, we agree with the reviewer that sustained exposure may not mimic completely the common use of antibiotics in children and other studies should be designed to test the most relevant and clinically translatable number and duration of repeated exposures of antibiotics, including recovery phases, during early life.

As suggested by the reviewer, this remark regarding the continuous exposure has been emphasized in the discussion section (page 15).

- 3) *Key information regarding stool sample collection and bioinformatics steps (e.g., which quality filtering thresholds, read-joining, and OTU clustering algorithms were used?) is missing.*

Response:

We have now changed the method section regarding the 16S rRNA gene sequence analysis in order to include more details about the technique, as asked by the reviewer. This paragraph has been moved to supplemental material.

- 4) *I would encourage the authors to deposit their sequence data into a public archive. There is no mention of sequence accession numbers in the manuscript.*

Response:

Sequences are now deposited in the QIITA website (<http://qiita.microbio.me>) under study ID 10774. This information has been added in the method section (and supplemental), following the description of the 16S rRNA gene sequence analysis.

- 5) *The authors used parametric statistics (repeated measures ANOVA) in their analysis of microbiome data over time. Did the data meet the assumptions of normality and equality of variance?*

Response:

When using the repeated measures ANOVA, we have to check the assumption of sphericity (which refers to the equality of variances of the differences between conditions). As a general rule, we use the Mauchly's test which tests the hypothesis that the variances of the differences between conditions are equal. If Mauchly's test statistic is not significant ($p > 0.05$), we conclude that the variances of differences are not significantly different. However, if Mauchly's test statistic is significant ($p < 0.05$), we conclude that there are significant differences between the variances and therefore the condition of sphericity is not met. In that case, we have to apply corrections to produce a valid F-ratio, like Greenhouse-Geisser correction. However, this assumption of sphericity is required only when we have a least 3 conditions which is the case for the dams analysis (within-subjects factor "time" with 3 levels: T0, T1, T2). Post-hoc tests were then analyzed by using Bonferonni correction. Regarding the analysis of microbiota over time in offspring, we had repeated-measures variable that has only 2 levels (time: PND21 and PND42) and therefore sphericity was automatically met and the F-ratio and degree of freedom (df) produced with Greenhouse-Geisser correction had exactly the same value as the non-corrected F-ratio and df when sphericity is assumed. An additional step was to check homogeneity of variances for each combination of the groups of the two factors (within-subjects factor (time) and between-subjects factor (treatment)). We tested it with the Levene's test. If homogeneity of variances was met, we then applied Bonferonni method for post-hoc tests which is the most robust to control Type I error. In some cases, we applied the Games-Howell procedure which is used when homogeneity of variances has not been assumed. Also, in a very few cases, significant outliers were identified graphically (with boxplot) and by using the "studentized residual" option of SPSS. The analysis was run with and without the outlier. Since the results were not statistically significant, the outliers were kept in the analysis. This approach has been added in the revised version of the manuscript (page 26).

- 6) *Pg 23, RNA and RT-qPCR methods – Please describe the genes (or at least types of genes) that were evaluated for differential expression.*

The list of genes and sequences measured by qPCR are described in the supplemental table 4. We used GAPDH as housekeeping gene after checking that the experimental treatments did not affect its expression in all tissues tested (ileum, colon, hippocampus and frontal cortex) (one-way ANOVA, $P > 0.05$) (see figure below. Results are means \pm SD). Experiments on

low-dose penicillin exposure in early life has also used GAPDH as an internal control for the qPCR analysis¹.

7) Pg 10, line 188 – Why was the unweighted UniFrac metric chosen to display this data, particularly in light of the discussion of the relative abundances of Proteobacteria that follow the reporting of the UniFrac-based results?

We thank the reviewer for pointing this out and we agree that analysis of weighted UniFrac is appropriate in this case. The difference of β -diversity between the treatment groups remain statistically significant in the weighted UniFrac analysis ($P = 0.001$). We have now added the weighted UniFrac analysis of dams and pups in the supplemental information (Fig S6 and S10) (page 9).

8) Figure 2 (and most others) --- please indicate the number of animals represented in each group and/or comparison.

Response:

The number of animals in each group and comparison has been now added in the legends of the figures.

9) Figure 3 – The same amount of variation is explained along the axes of the three PCoA plots. Given that the three plots are presented to illustrate independent points, they

should also be constrained specifically to the data contained within each (i.e., each PCoA should be specific to the data it contains, not sub-plotted from the analysis of data points contained in each of the three frames). This applies to Figure 4A as well.

We have now analyzed each time point separately and have replaced the figures in the main (Fig 3) text and in the supplemental data (Fig S5, S6 and S10).

10) Pg 12, line 241 – Increases in Avpr1b expression are described as being increased in the frontal cortex but not hippocampus. The corresponding figure (Fig 6A/B) does not include the hippocampal data.

Response:

We have included the data related to the hippocampal expression of Avpr1b in males and females in supplemental figure 17. The results indicate no statistical difference between AB-treated mice and controls.

References:

1. Cox, L. M. *et al.* Altering the intestinal microbiota during a critical developmental window has lasting metabolic consequences. *Cell* **158**, 705–721 (2014).
2. Russell, S. L. *et al.* Early life antibiotic-driven changes in microbiota enhance susceptibility to allergic asthma. *EMBO Rep.* **13**, 440–447 (2012).
3. Russell, S. L. *et al.* Perinatal antibiotic-induced shifts in gut microbiota have differential effects on inflammatory lung diseases. *Journal of Allergy and Clinical Immunology*. 135, 100-109.e5 (2015)

REVIEWERS' COMMENTS:

Reviewer #1 (Remarks to the Author):

I would like to thank the authors for their careful consideration of the reviewer comments in producing this substantially revised manuscript. I agree that future studies investigating the minimum number and duration of repeated AB exposures in early life that lead to long-term changes in host physiology, the importance of timing of treatment in the development of a specific adult phenotype and the effects of JB-1 alone will yield important insights. I have one further recommendation for the authors.

They now state in the revised paper that they found *L. rhamnosus* was detectable in feces of all dams before treatment and only in the dams of the AB/JB1 group after treatment but this data is not shown. As the authors note, this is important to prove that the protocol is 'efficient in maintaining the presence of some *L. rhamnosus* in the gut' and should be included in the manuscript.

Reviewer #2 (Remarks to the Author):

Thank you for the very thorough response to the reviewer comments. I have not other comments or suggestions.

Reviewer #3 (Remarks to the Author):

In this revision, the authors have addressed many of the concerns that I raised with their initial submission. However, there are a few lingering issues that should be addressed. These include: 1) Number of animals used in each test and/or per treatment. Although this has been addressed in most of the figures now, it is not displayed in Figure S5.

2) What statistic was used to compare beta-diversity? Results from the evaluation of beta diversity levels among groups appear in Figures S5, S6, and S10, but neither the main or supplemental methods describe the test(s) used.

3) Specifically, how were chimeric sequences identified? What algorithm was used?

4) What was done with the core OTUs? Their identification is mentioned in the supplemental methods, but it is not apparent what, if anything, was done with this data downstream.

5) In Supplemental Table 1, please indicate if these are mean values and provide an estimate of variance (e.g., standard deviation, IQR, SEM) to go along with them.

6) Figure 4 – Although I assume that the same order (of presentation) and color scheme are used in this figure (as in others) to depict treatment groups, please be sure that they are defined here as well.

7) Throughout the manuscript, please verify that manufacturer and/or publisher information is provided for the tools and materials used to carry out the study. E.g., pg 25, line 547 "Image J". Several instances of missing manufacturer/publisher information also appear in the supplemental methods.

8) Throughout the manuscript, please double check noun/verb agreement, particularly in the newly

added sections of text.

9)Pg 16, line 329 – please provide additional context for the statement regarding lipopolysaccharides. While the meaning of this may be apparent to a microbiologist or immunologist, this may not be the case for readers of other backgrounds.

10)Pg 19, line 407-408 – Although the window antibiotic exposure overlaps with the known developmental window for TJ protein formation, it seems inappropriate to claim that the exact “moment” was captured.

Point-by-point response to referees 'comments

Reviewer#1:

*They now state in the revised paper that they found *L. rhamnosus* was detectable in feces of all dams before treatment and only in the dams of the AB/JB1 group after treatment but this data is not shown. As the authors note, this is important to prove that the protocol is 'efficient in maintaining the presence of some *L. rhamnosus* in the gut' and should be included in the manuscript*

We have now added the figure of the quantification by qPCR of *Lactobacillus rhamnosus* in the feces of dams (Supplementary Figure 20) (page 23, method section, *Treatment with antibiotic and L. rhamnosus JB-1TM*)

Reviewer#3:

- 1) *Number of animals used in each test and/or per treatment. Although this has been addressed in most of the figures now, it is not displayed in Figure S5.*

The number of animals has now been added in Supplementary Figure 5

- 2) *What statistic was used to compare beta-diversity? Results from the evaluation of beta diversity levels among groups appear in Figures S5, S6, and S10, but neither the main or supplemental methods describe the test(s) used.*

We have now added to the methods that the significance between groups for distances was assessed using t-tests as implemented in "make_distance_boxplots.py" script in QIIME 1.8.0

- 3) *Specifically, how were chimeric sequences identified? What algorithm was used?*

Chimeric sequences were identified using USEARCH and this has now been added to the text.

4) *What was done with the core OTUs? Their identification is mentioned in the supplemental methods, but it is not apparent what, if anything, was done with this data downstream.*

We have now made it clearer in the methods that the downstream analysis was carried out only on the core OTUS.

5) *In Supplemental Table 1, please indicate if these are mean values and provide an estimate of variance (e.g., standard deviation, IQR, SEM) to go along with them.*

We have now added the standard deviations that go along with the mean values and added this information in the legend.

6) *Figure 4 – Although I assume that the same order (of presentation) and color scheme are used in this figure (as in others) to depict treatment groups, please be sure that they are defined here as well.*

We thank the reviewer for his/her careful attention. We have now added the legend of group in Figure 4.

7) *Throughout the manuscript, please verify that manufacturer and/or publisher information is provided for the tools and materials used to carry out the study. E.g., pg 25, line 547 “Image J”. Several instances of missing manufacturer/publisher information also appear in the supplemental methods.*

The reference Schneider et al, Nature Methods, 2012 has been added in regard with the use of Image J, and the origin of product have been added in the method section.

8) *Throughout the manuscript, please double check noun/verb agreement, particularly in the newly added sections of text.*

Check.

9) *Pg 16, line 329 – please provide additional context for the statement regarding lipopolysaccharides. While the meaning of this may be apparent to a microbiologist or immunologist, this may not be the case for readers of other backgrounds*

We have added, in page 16, that lipopolysaccharides are endotoxins that can elicit strong immune responses in the host.

10) *Pg 19, line 407-408 – Although the window antibiotic exposure overlaps with the known developmental window for TJ protein formation, it seems inappropriate to claim that the exact “moment” was captured.*

We have changed the sentence according to the remark of the reviewer, as follows: “In our study, AB exposure of fetus started at E12 – E14, a period which overlaps with the developmental window of TJ protein formation.”